EMBO
Molecular Medicine

# Dextrorotatory kynurenine suppresses acute rejection through inhibiting M1 macrophage-mediated inflammation

Yufeng Zhao[1,2,3,6], Jiaheng Wu [2,3,6], Yuling Li[4,6], Yirui Cao[2], Tongyu Zhu[2,3], Yinlong Guo [4✉], Cheng Yang [2,3,5✉] & Dong Zhu [1,2,3✉]

## Abstract

Acute rejection (AR) remains a critical challenge to graft survival in kidney transplantation. Although dextrorotatory-amino acids (D-AAs) have been recognized as biologically active compounds, their role in mediating immunosuppression was poorly depicted. To address this, serum samples from renal transplant recipients were analyzed via [d0]/[d5]-estradiol-3-benzoate-17β-chloroformate (17β-EBC) based ion mobility-mass spectrometry (IM-MS) to assess D-AAs levels. scRNA-seq data from the GSE109564 dataset were analyzed. Additionally, murine skin and kidney transplantation models were utilized to assess the in vivo impact of d-kynurenine (D-Kyn) treatment on AR. Through analysis of patient serum and murine transplantation models, we identified D-Kyn as a key metabolite whose elevated levels correlate with stable graft function. We found that D-Kyn, more effectively than its chiral counterpart L-Kyn, inhibits the inflammatory activity of M1 macrophages. This suppression is mediated via the PHGDH/TLR4/Caspase-1 pathway, reducing the transcription and secretion of inflammatory cytokines. In murine models of skin and kidney transplantation, D-Kyn treatment demonstrated potent immunosuppressive effects, attenuating macrophage-mediated inflammation and CD8 + T cell activation, potentially through regulation of macrophage-derived IL-23a. Our findings reveal D-Kyn as a promising therapeutic candidate for preventing acute rejection and improving transplant outcomes and lay the foundation for future clinical applications from the perspective of dextrorotatory amino acids.

**Keywords** Renal Transplantation; Acute Rejection; D-Kyn; Macrophages Modulation; Enantiomeric Analysis
**Subject Category** Immunology

## Introduction

Renal transplantation remains the optimum treatment for patients suffering from end-stage renal disease (ESRD). Despite substantial advances in immunological compatibility and immunosuppressive therapies, acute rejection (AR) continues to pose a considerable challenge, contributing to long-term graft loss (Hart et al, 2017). AR is a complex pathological process involving activation of complement and various immune cells, including macrophages and T cells, which orchestrate an inflammatory response that damages the graft (Bouatou et al, 2019; Nguan, 2013).

Amino acids (AAs) are essential biological molecules that are increasingly recognized for their role in diverse immune modulation and disease pathogenesis (Chen et al, 2022; Proietti et al, 2020). Recent advances in stereospecific recognition have highlighted that D-AAs prove not to be merely structural variants but are also physiologically active compounds with significant immunosuppressive effects. D-AAs have been showed to influence the viability of immune cells, including those involved in graft rejection (Du et al, 2020). For instance, D-kynurenine (D-Kyn), a metabolite derived from Trp, has been found to inhibit T cell proliferation and promote apoptosis, thereby exhibiting immunosuppressive properties both in vitro and in vivo. The aryl hydrocarbon receptor (AHR) signaling pathway mediates the immunomodulatory effects of Kyn (Holmgaard et al, 2015; Munn et al, 2005), particularly by enhancing the expression of anti-inflammatory genes in macrophages (Memari et al, 2015). These properties suggest that D-Kyn and other D-AAs may play a crucial role in modulating the immune response in AR, potentially improving transplant outcomes by influencing key immune cells such as macrophages and T cells.

Macrophages play a pivotal role in mediating the immune response during AR, with the M1 phenotype of macrophages being especially influential (Ordikhani et al, 2020). In addition to their role in innate immunity, macrophages are well recognized for their capacity to initiate adaptive immune responses, proving to be instrumental in perpetuating acute rejection (Li et al, 2019). In the context of renal transplantation, macrophages often display a pro-

[1]Department of Kidney Transplantation, Zhongshan Hospital, Fudan University, Shanghai, China. [2]Shanghai Key Laboratory of Organ Transplantation, Zhongshan Hospital, Fudan University, Shanghai, China. [3]Department of Urology, Zhongshan Hospital (Xiamen), Fudan University, Xiamen, Fujian, China. [4]National Center for Organic Mass Spectrometry in Shanghai, Shanghai Institute of Organic Chemistry, Chinese Academy of Sciences, Shanghai, China. [5]Zhangjiang Institute of Fudan University, Shanghai, China. [6]These authors contributed equally: Yufeng Zhao, Jiaheng Wu, Yuling Li. ✉E-mail: ylguo@sioc.ac.cn; esuperyc@163.com; zhu.dong@zs-hospital.sh.cn

inflammatory phenotype, secreting an array of cytokines like IL-23a, that activate endothelial cells and facilitate the activation of cytotoxic T cells (Mueller et al, 2019). This underscores the profound influence of macrophages in the rejection process. While macrophages are recognized as key players in driving AR, the

specific mechanisms by which Kyn regulates their activity remain incompletely understood. Additionally, the differential roles of D-Kyn and its isomer L-Kyn in modulating macrophage activity are not well-defined, representing a critical gap in our understanding. Addressing these mechanistic and chirality gaps is essential for advancing our knowledge of immune regulation in AR.

In the present study, we aimed to investigate how D-Kyn modulates macrophage activity, particularly focusing on its effects on IL-23a production and subsequent CD8 T cell activation during AR. To achieve this, the novel [d0]/[d5]-estradiol-3-benzoate-17β-chloroformate (17β-EBC) based ion mobility-mass spectrometry (IM-MS) was applied to compare the D-/L-AAs ratio in whole blood samples of recipients with AR and those with stable kidney function. This approach enabled us to identify potential chiral biomarkers for the diagnosis of AR. Mechanistically, D-Kyn demonstrated the ability to suppress M1 macrophage activity by inhibiting phosphoglycerate dehydrogenase (PHGDH) via the ERK1/2/FOS pathway, outperforming its chiral counterpart, L-Kyn. Additionally, D-Kyn exhibited greater potency in inhibiting the transcription and secretion of inflammatory cytokines through the PHGDH/TLR4/Caspase-1 pathway. In both murine skin and kidney transplantation models, D-Kyn outperformed the immunosuppressive potency of L-Kyn. D-Kyn administration reduced

**Table 1.** Clinical profile comparison of renal allograft recipients in the acute rejection and stable groups.

|  | AR ($n = 12$) | Stable ($n = 11$) | P value |
|---|---|---|---|
| Gender (male), n (%) | 9 (75.0%) | 9 (81.8%) | 1.00 |
| Age, years | 50 ± 11 | 47 ± 14 | 0.74 |
| HLA mismatch, n (%) |  |  | 0.75 |
| 1 | 1 (8.3%) | 2 (18.2%) |  |
| 2 | 5 (41.7%) | 5 (45.5%) |  |
| 3 | 4 (33.3%) | 2 (18.2%) |  |
| 4 | 2 (16.7%) | 1 (9.1%) |  |
| 5 | 0 (0%) | 1 (9.1%) |  |
| 6 | 0 (0%) | 0 (0%) |  |
| Serum creatine (μmol/L) | 224.0 ± 70.7 | 155.3 ± 53.4 | 0.01 |

HLA human leukocyte antigen.

**Table 2.** The overall illustration of the clinical background of 23 cases of kidney transplant recipients.

| Recipients | Gender | Age | PRA I | PRA II | Immunosuppressant regiments | Post-operation time | Latest Scr (μmol/L) |
|---|---|---|---|---|---|---|---|
| AR1 | F | 69 | 30 | 0 | CsA + MMF + Pred | 1 m | 213 |
| AR2 | M | 48 | 0 | 0 | Tac + MMF + Pred | 3 y | 271 |
| AR3 | M | 59 | 0 | 69 | CsA + MMF + Pred | 2 y | 316 |
| AR4 | F | 55 | 0 | 17 | CsA + MMF + Pred | 3w | 155 |
| AR5 | M | 44 | 0 | 51 | Rapa + MMF + Pred | 15 y | 147 |
| AR6 | M | 53 | 0 | 29 | Tac + MMF + Pred | 1 y | 181 |
| AR7 | F | 53 | 0 | 0 | CsA + MMF + Pred | 13 y | 372 |
| AR8 | M | 47 | 0 | 0 | Tac + MMF + Pred | 14 m | 197 |
| AR9 | M | 39 | 0 | 0 | Tac + MMF + Pred | 4 y | 232 |
| AR10 | M | 54 | 0 | 0 | Tac + MMF + Pred | 9 m | 139 |
| AR11 | M | 57 | 0 | 54 | CsA + MMF + Pred | 9 m | 205 |
| AR12 | M | 24 | 0 | 0 | Rapa + MMF + Pred | 7 y | 260 |
| S1 | M | 57 | 0 | 71 | CsA + MMF + Pred | 1 m | 130 |
| S2 | F | 63 | 0 | 0 | Tac + MMF + Pred | 1 m | 62 |
| S3 | M | 30 | 0 | 0 | Tac + MMF + Pred | 1 m | 132 |
| S4 | M | 39 | 0 | 0 | Tac + MMF + Pred | 1 m | 110 |
| S5 | M | 64 | 0 | 0 | CsA + MMF + Pred | 1 m | 167 |
| S6 | M | 34 | 0 | 0 | Tac + MMF + Pred | 2 y | 149 |
| S7 | M | 26 | 0 | 0 | Tac + MMF + Pred | 1w | 140 |
| S8 | M | 51 | 0 | 0 | Tac + MMF + Pred | 1w | 274 |
| S9 | M | 46 | 0 | 6 | Tac + MMF + Pred | 14 y | 185 |
| S10 | M | 49 | 0 | 23 | Tac + MMF + Pred | 18 y | 180 |
| S11 | F | 62 | 0 | 0 | Tac + MMF + Pred | 5 m | 179 |

AR acute rejection, S stable kidney function, CsA cyclosporin A, Tac tacrolimus, MMF mycophenolate mofetil, Rapa rapamycin, Pred prednisone, Scr serum creatinine.

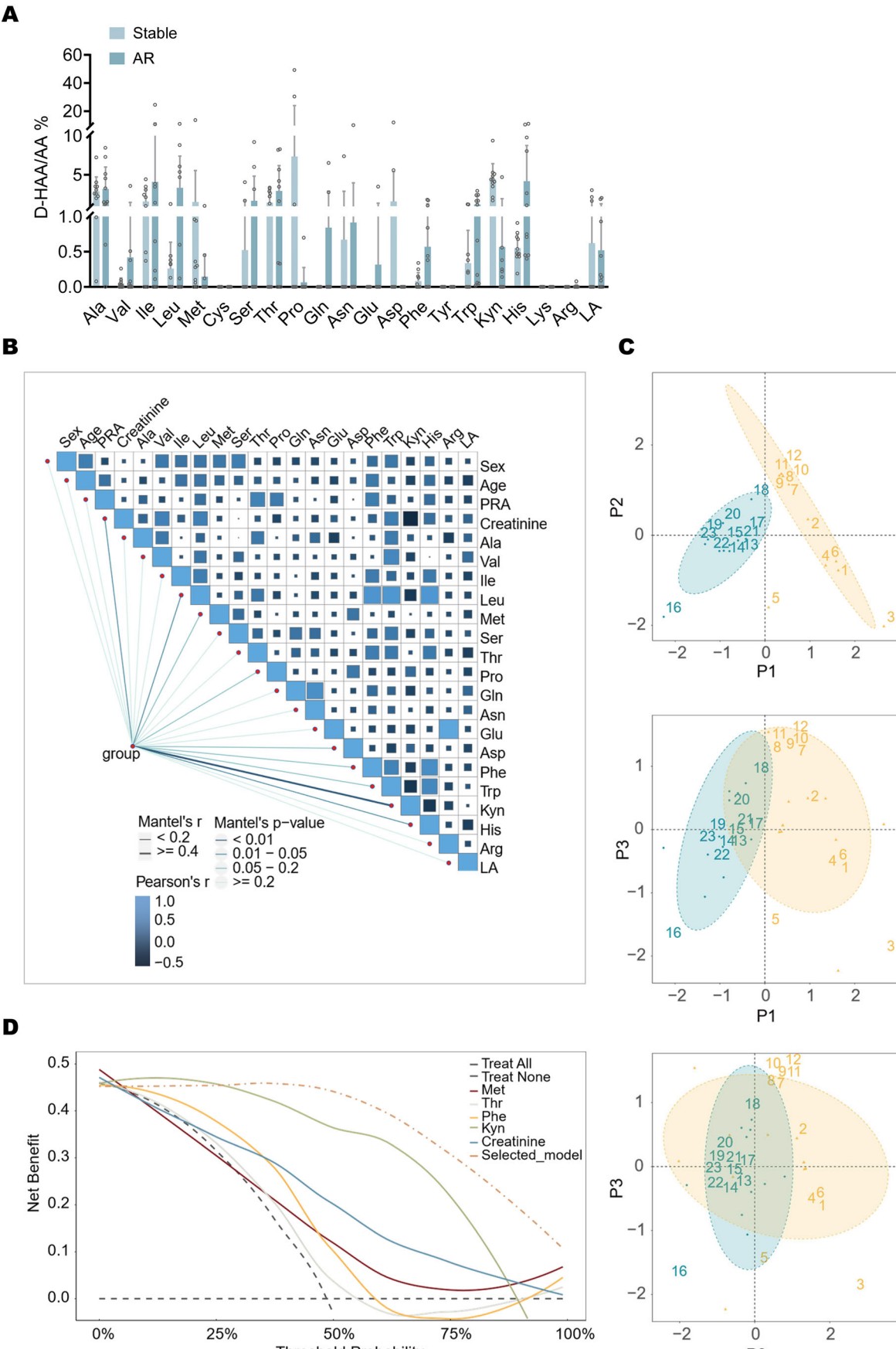

◀ **Figure 1. Comprehensive dextrorotatory profiling of 20 AAs and correlation analysis in renal transplantation.**

(A) The overall demonstration of the ratio of D-AA/total-AA of 20 AAs and one metabolite, LA, detected by the MS approach. (B) The linkET plot showing the correlation between each detected AA and clinical factor using the Pearson algorithm and the correlation between the overall grouping and different factors using the Mantel algorithm. (C) The demonstration of the separation of AR and stable patients using our D-AAs-based diagnosing model. The different components were stimulated using the PLS algorithm. (D) The DCA results show a higher overall net benefit of our model, compared to individual D-AA. Source data are available online for this figure.

**Table 3. The proportion of D-AA/total-AA of 20 AAs in the stable group assessed by the MS approach.**

|  | S1 | S2 | S3 | S4 | S5 | S6 | S7 | S8 | S9 | S10 | S11 | Mean | SD |
|---|---|---|---|---|---|---|---|---|---|---|---|---|---|
| Ala | 3.06% | 2.24% | 7.23% | 2.01% | 0.99% | 4.03% | 2.59% | 3.51% | 0.08% | 2.54% | 3.49% | 2.89% | 1.84% |
| Val | 0.00% | 0.07% | 0.26% | 0.10% | 0.00% | 0.00% | 0.03% | 0.03% | 0.04% | 0.00% | 0.00% | 0.05% | 0.08% |
| Ile | 0.37% | 3.48% | 0.49% | 2.43% | 4.42% | 0.00% | 0.00% | 0.00% | 2.14% | 3.07% | 1.02% | 1.58% | 1.59% |
| Leu | 0.00% | 0.00% | 0.32% | 1.26% | 0.33% | 0.00% | 0.00% | 0.00% | 0.13% | 0.38% | 0.44% | 0.26% | 0.37% |
| Met | 0.96% | 0.28% | 0.36% | 0.94% | 0.30% | 0.00% | 0.00% | 0.00% | 0.06% | 0.10% | 13.66% | 1.51% | 4.04% |
| Cys | 0.00% | 0.00% | 0.00% | 0.00% | 0.00% | 0.00% | 0.00% | 0.00% | 0.00% | 0.00% | 0.00% | 0.00% | 0.00% |
| Ser | 0.00% | 0.00% | 0.00% | 1.69% | 0.00% | 0.00% | 0.00% | 0.00% | 0.00% | 0.00% | 4.05% | 0.52% | 1.28% |
| Thr | 3.23% | 0.00% | 2.63% | 2.23% | 3.36% | 0.00% | 0.00% | 0.00% | 2.87% | 1.33% | 0.00% | 1.42% | 1.46% |
| Pro | 49.25% | 0.00% | 0.00% | 1.20% | 0.00% | 0.00% | 0.00% | 0.00% | 0.00% | 30.32% | 0.00% | 7.34% | 16.59% |
| Gln | 0.00% | 0.00% | 0.00% | 0.00% | 0.00% | 0.00% | 0.00% | 0.00% | 0.00% | 0.00% | 0.00% | 0.00% | 0.00% |
| Asn | 0.00% | 0.00% | 7.38% | 0.00% | 0.00% | 0.00% | 0.00% | 0.00% | 0.00% | 0.00% | 0.00% | 0.67% | 2.22% |
| Glu | 0.00% | 0.00% | 0.00% | 0.00% | 0.00% | 0.00% | 0.00% | 0.00% | 0.00% | 0.00% | 0.00% | 0.00% | 0.00% |
| Asp | 0.00% | 0.00% | 0.00% | 0.00% | 0.00% | 0.00% | 0.00% | 0.00% | 0.00% | 11.97% | 5.59% | 1.60% | 3.83% |
| Phe | 0.00% | 0.34% | 0.00% | 0.25% | 0.00% | 0.00% | 0.04% | 0.00% | 0.00% | 0.13% | 0.15% | 0.08% | 0.12% |
| Tyr | 0.00% | 0.00% | 0.00% | 0.00% | 0.00% | 0.00% | 0.00% | 0.00% | 0.00% | 0.00% | 0.00% | 0.00% | 0.00% |
| Trp | 0.46% | 0.22% | 1.21% | 0.38% | 0.00% | 1.24% | 0.00% | 0.00% | 0.00% | 0.00% | 0.20% | 0.34% | 0.47% |
| Kyn | 4.42% | 4.84% | 4.40% | 9.37% | 3.50% | 2.11% | 5.28% | 4.11% | 3.96% | 5.10% | 1.71% | 4.44% | 1.99% |
| His | 0.71% | 0.47% | 0.91% | 0.68% | 0.54% | 0.19% | 0.46% | 0.48% | 0.43% | 0.43% | 0.77% | 0.55% | 0.20% |
| Lys | 0.00% | 0.00% | 0.00% | 0.00% | 0.00% | 0.00% | 0.00% | 0.00% | 0.00% | 0.00% | 0.00% | 0.00% | 0.00% |
| Arg | 0.00% | 0.00% | 0.00% | 0.00% | 0.00% | 0.00% | 0.00% | 0.00% | 0.00% | 0.00% | 0.00% | 0.00% | 0.00% |
| LA | 0.00% | 0.00% | 0.00% | 0.00% | 0.00% | 1.65% | 3.03% | 2.15% | 0.00% | 0.00% | 0.00% | 0.62% | 1.11% |

macrophage-mediated inflammation and CD8 T cell activation, potentially by regulating IL-23a. These results suggest that D-Kyn could be a therapeutic target for AR, bridging the knowledge gap regarding its role in modulating immune responses during AR.

## Results

### Clinical profiling of the renal transplant patients

A total of 23 cases, comprising 12 AR patients and 11 stable post-transplantation patients, were enrolled from a total of 25 patients. Comparative analysis of several factors, including age, gender, HLA typing, immunosuppressive regime and kidney source, revealed no significant differences, as shown in Table 1. Detailed information on the 23 patients is provided in Table 2. Two patients were excluded on the grounds of drug-induced renal injury verified by pathological examination and pre-renal injury caused by low circulating blood volume. Seven patients (5 AR/2 stable) were subjected to immunosuppressive therapy with a combination of cyclosporin (CsA) + mycophenolate mofetil (MMF) + prednisone

(Pred), while 14 cases (5 AR/9 stable) received tacrolimus (FK) + MMF + Pred. The rest 2 patients (both AR) were under the prescription of rapamycin (Rapa) + MMF + Pred.

### Enantio-analysis and correlation analysis of AAs in renal transplantation

Utilizing the novel ion mobility-mass spectrometry-based approach, we analyzed the proportion of D-AAs/total-AAs for 20 AAs and lactic acid (LA) in serum samples (Fig. 1A and Tables 3 and 4). Significant differences in dextrorotatory composition were observed for Val, Leu, Phe, His, Trp, and Kyn between the two groups (Table 5). Markedly higher proportion of D-Kyn was found in the serum of stable patients, while D-Val, D-Leu, D-Phe, and D-Trp were up-regulated in AR patients. Correlation analysis revealed strong associations between D-Leu, D-His, D-Trp, and D-Phe, particularly in the AR group. Kyn showed the highest correlation with the separation of AR and stable group (Fig. 1B).

LASSO analysis was conducted to identify AAs and clinical factors with predictive values for AR (Fig. EV1A), which led to the selection of 4 AAs: Met, Thr, Phe, and Kyn. The optimal number of

**Table 4.** The proportion of D-AA/total-AA of 20 AAs in the AR group assessed by the MS approach.

|      | S1     | S2     | S3     | S4    | S5    | S6     | S7    | S8    | S9    | S10   | S11   | Mean  | SD    |
|------|--------|--------|--------|-------|-------|--------|-------|-------|-------|-------|-------|-------|-------|
| Ala  | 1.51%  | 5.10%  | 4.53%  | 1.23% | 8.45% | 3.27%  | 7.29% | 1.35% | 0.00% | 0.60% | 1.55% | 1.42% | 3.02% |
| Val  | 0.08%  | 0.02%  | 0.38%  | 0.51% | 0.04% | 0.00%  | 3.57% | 0.00% | 0.00% | 0.00% | 0.00% | 0.00% | 0.38% |
| Ile  | 24.47% | 10.66% | 0.00%  | 0.00% | 3.46% | 0.00%  | 3.84% | 0.11% | 0.23% | 0.65% | 1.43% | 2.20% | 3.92% |
| Leu  | 8.80%  | 4.68%  | 11.00% | 5.36% | 0.60% | 0.00%  | 6.06% | 0.00% | 0.00% | 0.00% | 0.12% | 0.00% | 3.05% |
| Met  | 0.12%  | 0.00%  | 0.45%  | 1.02% | 0.00% | 0.00%  | 0.00% | 0.00% | 0.00% | 0.00% | 0.00% | 0.00% | 0.13% |
| Cys  | 0.00%  | 0.00%  | 0.00%  | 0.00% | 0.00% | 0.00%  | 0.00% | 0.00% | 0.00% | 0.00% | 0.00% | 0.00% | 0.00% |
| Ser  | 0.00%  | 0.00%  | 0.00%  | 9.19% | 0.00% | 6.01%  | 3.15% | 0.00% | 0.00% | 0.00% | 0.00% | 0.00% | 1.53% |
| Thr  | 0.00%  | 3.46%  | 8.27%  | 5.66% | 8.18% | 4.20%  | 1.62% | 0.34% | 0.33% | 0.00% | 0.00% | 0.00% | 2.67% |
| Pro  | 0.00%  | 0.00%  | 0.00%  | 0.00% | 0.00% | 0.70%  | 0.00% | 0.00% | 0.00% | 0.00% | 0.00% | 0.00% | 0.06% |
| Gln  | 0.00%  | 2.75%  | 0.00%  | 0.00% | 0.00% | 6.52%  | 0.00% | 0.00% | 0.00% | 0.00% | 0.00% | 0.00% | 0.77% |
| Asn  | 0.00%  | 0.00%  | 0.00%  | 0.00% | 0.00% | 10.08% | 0.00% | 0.00% | 0.00% | 0.00% | 0.00% | 0.00% | 0.84% |
| Glu  | 0.00%  | 0.00%  | 0.00%  | 0.00% | 0.00% | 0.00%  | 0.00% | 0.00% | 3.48% | 0.00% | 0.00% | 0.00% | 0.29% |
| Asp  | 0.00%  | 0.00%  | 0.00%  | 0.00% | 0.00% | 0.00%  | 0.00% | 0.00% | 0.00% | 0.00% | 0.00% | 0.00% | 0.00% |
| Phe  | 1.73%  | 0.38%  | 1.82%  | 0.84% | 0.44% | 1.07%  | 0.00% | 0.00% | 0.00% | 0.00% | 0.00% | 0.00% | 0.52% |
| Tyr  | 0.00%  | 0.00%  | 0.00%  | 0.00% | 0.00% | 0.00%  | 0.00% | 0.00% | 0.00% | 0.00% | 0.00% | 0.00% | 0.00% |
| Trp  | 1.66%  | 1.67%  | 2.39%  | 2.42% | 0.66% | 1.18%  | 2.94% | 0.02% | 0.06% | 0.05% | 0.17% | 0.72% | 1.16% |
| Kyn  | 0.38%  | 0.22%  | 0.00%  | 0.56% | 4.69% | 0.25%  | 0.00% | 0.00% | 0.00% | 0.00% | 0.14% | 0.00% | 0.52% |
| His  | 8.03%  | 11.08% | 10.48% | 9.78% | 1.10% | 0.45%  | 0.40% | 0.71% | 0.75% | 0.45% | 2.63% | 1.02% | 3.91% |
| Lys  | 0.00%  | 0.00%  | 0.00%  | 0.00% | 0.00% | 0.00%  | 0.00% | 0.00% | 0.00% | 0.00% | 0.00% | 0.00% | 0.00% |
| Arg  | 0.00%  | 0.00%  | 0.00%  | 0.00% | 0.00% | 0.00%  | 0.00% | 0.00% | 0.08% | 0.00% | 0.00% | 0.00% | 0.01% |
| LA   | 0.14%  | 0.00%  | 0.00%  | 0.00% | 0.09% | 1.90%  | 2.04% | 0.00% | 0.17% | 0.44% | 0.93% | 0.20% | 0.49% |

**Table 5.** The differences in the percentage of dextrorotatory composition of 20 AAs, one metabolite, LA and level of Scr.

|      | AR          | Stable      | P value |
|------|-------------|-------------|---------|
| Ala  | 0.03 ± 0.03 | 0.03 ± 0.02 | 0.891   |
| Val  | 0.00 ± 0.01 | 0.00 ± 0.00 | 0.289   |
| Ile  | 0.04 ± 0.07 | 0.02 ± 0.02 | 0.302   |
| Leu  | 0.03 ± 0.04 | 0.00 ± 0.00 | 0.031   |
| Met  | 0.00 ± 0.00 | 0.02 ± 0.04 | 0.251   |
| Cys  | 0.00 ± 0.00 | 0.00 ± 0.00 | NA      |
| Ser  | 0.02 ± 0.03 | 0.01 ± 0.01 | 0.322   |
| Thr  | 0.03 ± 0.03 | 0.01 ± 0.01 | 0.252   |
| Pro  | 0.00 ± 0.00 | 0.07 ± 0.17 | 0.142   |
| Gln  | 0.01 ± 0.02 | 0.00 ± 0.00 | 0.21    |
| Asn  | 0.01 ± 0.03 | 0.01 ± 0.02 | 0.878   |
| Glu  | 0.00 ± 0.01 | 0.00 ± 0.00 | 0.35    |
| Asp  | 0.00 ± 0.00 | 0.02 ± 0.04 | 0.162   |
| Phe  | 0.01 ± 0.01 | 0.00 ± 0.00 | 0.049   |
| Tyr  | 0.00 ± 0.00 | 0.00 ± 0.00 | NA      |
| Trp  | 0.01 ± 0.01 | 0.00 ± 0.00 | 0.026   |
| Kyn  | 0.01 ± 0.01 | 0.04 ± 0.02 | <0.001  |
| His  | 0.04 ± 0.04 | 0.01 ± 0.00 | 0.022   |
| Lys  | 0.00 ± 0.00 | 0.00 ± 0.00 | NA      |
| Arg  | 0.00 ± 0.00 | 0.00 ± 0.00 | 0.35    |
| LA   | 0.00 ± 0.01 | 0.01 ± 0.01 | 0.747   |

components for the model was determined by assessing the root mean square error of prediction (RMSEP), with the lowest RMSEP observed when the number of components was set to one, followed by two and then three components (Fig. EV1B). Considering the variance explained (VE), a model with three components was chosen for further PLS analysis. The PLS analysis effectively distinguished between patients with AR and stable kidney function (Fig. 1C). The accuracy of our predictive model was validated using DCA (Fig. 1D). Moreover, a nomogram model was developed to assess the risk of AR incidence based on the individual levels of each D-AA (Fig. EV1C).

## Identification of potential correlation between Kyn and M1 macrophage in AR

Given the emerging role of M1 macrophages in mediating AR (Schmauch et al, 2024; Yu and Lu, 2022), we assessed the role of macrophages through analyzing the scRNA-seq data of AR sample biopsy in the GSE109564 dataset. A total of 16 subclusters were re-clustered and annotated (Figs. 2A,B and EV2). All the marker gene expression for each cluster was depicted in Fig. EV3. Two macrophage clusters (Mac-1 and Mac-2) were identified in AR kidneys using data from the GSE109564 database. Mac-1, characterized by high levels of inflammatory mediators such as IL-1β and CD83 (Ma et al, 2021), closely resembled M1 inflammatory macrophages (Fig. EV3). A significant discrepancy in the expression of kynureninase (KYNU), a key enzyme directly involved in the catabolism of kynurenine, and IL-1β was observed between these two macrophage subtypes (Fig. 2C). Pseudo-time

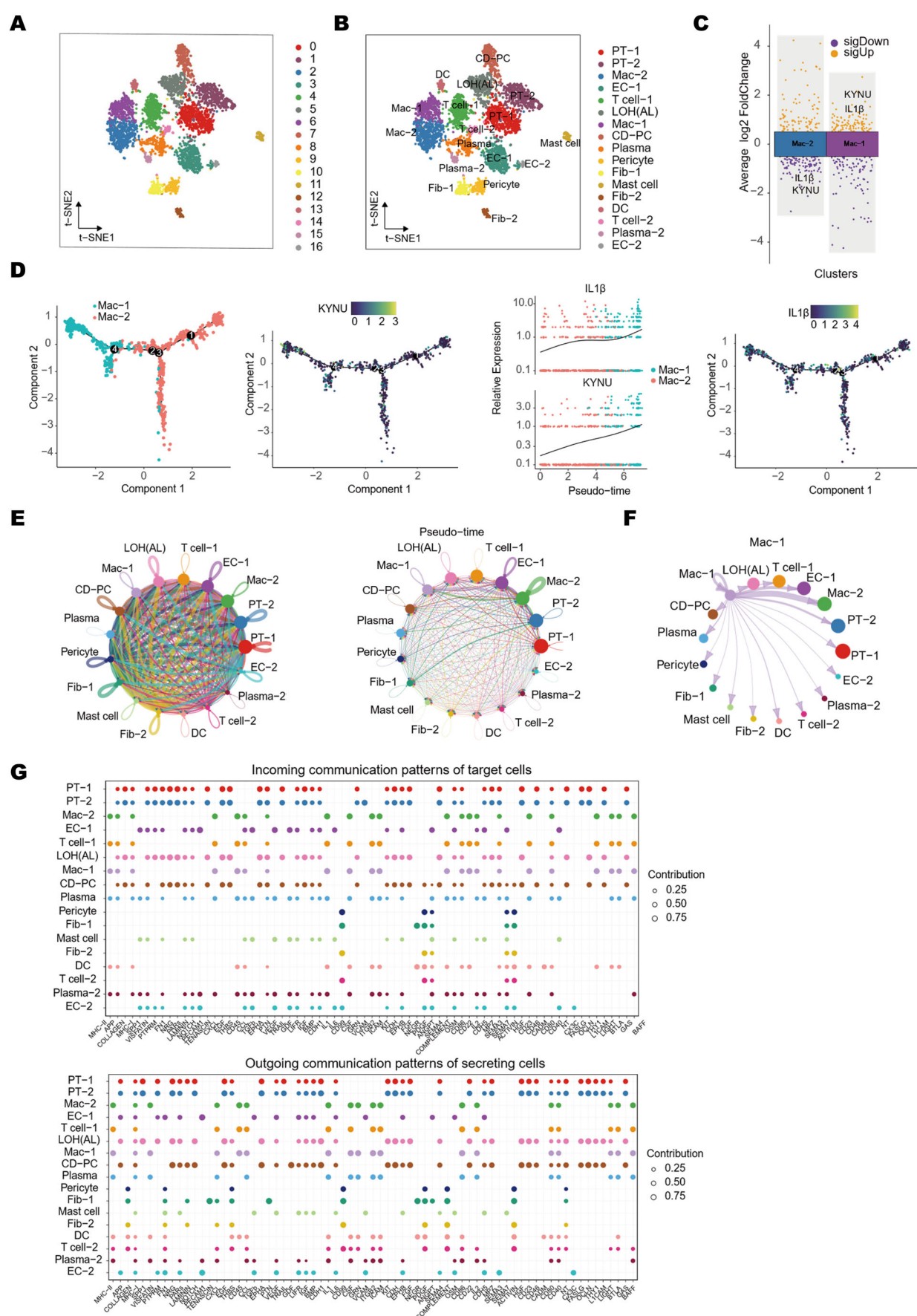

**Figure 2.   Identification of a potential correlation between KYN and M1 macrophage in AR.**

(A, B) scRNA-seq analysis of kidney biopsy samples undergoing rejection yielding a total of 16 distinct subclusters (A) and annotated subcluster illustration (B). (C) The expression differences in 2 distinct subclusters of macrophages in the AR kidneys, with IL-1β and KYNU highlighted. (D) Pseudo-time analysis illustrated divergent differentiation trajectories for Mac-1 and Mac-2, with a special focus on the shift of IL-1β and KYNU expression with the pseudo-time progression. (E) Overall illustration of cell-to-cell communication between the 16 annotated main cell clusters in the AR kidney. (F) Overall illustration of cell-to-cell communication of Mac-1 in AR kidney. (G) The incoming and outgoing communication analysis of 16 main cell types annotated in the AR kidney. Source data are available online for this figure.

analysis further highlighted distinct differences in the differentiation trajectories of Mac-1 and Mac-2 (Fig. 2D). Analyzing the expression of KYNU and IL-1β along these pseudo-temporal trajectories revealed a marked positive correlation between the two as pseudo-time progressed. Cellchat analysis was employed to explore cellular communication, revealing active interaction among macrophages, proximal tubular cells, T and endothelial cells in AR (Fig. 2E). Mac-1 was found to interact with Mac-2 and endothelial cells (Fig. 2F). The main incoming signals to Mac-1 was identified, including pathways related to chemokines and antigen-presenting molecules (Fig. 2G). Outgoing signaling from Mac-1 were enriched in various inflammatory pathways, such as IL-1, IL-2, and CD86. Collectively, these results suggest that Mac-1 macrophage interacted with various cell types primarily through inflammatory pathways in the context of allograft rejection. Additionally, Mac-1 may preferentially utilize Kyn under the AR-triggered immune microenvironment.

## D-Kyn displays higher potency in suppressing inflammatory secretion of M1 macrophages than L-Kyn

D-Kyn was identified to be consistently up-regulated in the non-acute rejection group and displayed the highest correlation with the separation of AR and stable group. Kyn has been established to regulate immunosuppressive responses through AHR signaling pathway (Holmgaard et al, 2015; Munn et al, 2005), which actively promotes expression of anti-inflammatory genes in macrophages (Memari et al, 2015). This led us hypothesize that D-Kyn might be functionally linked to the pathogenesis of AR through regulating macrophages. We cultured and induced the differentiation of M1 from THP-1 and RAW264.7 cell lines, which were further subjected to D-Kyn and L-Kyn treatment. A down-regulation of key M1 markers in mRNA levels, particularly IL-1β and IL-6, was observed in M1 macrophages treated with both forms of Kyn (Fig. 3A,B), indicating significant suppression of M1-mediated inflammation. Intriguingly, we observed significantly lower levels of protein secretion of IL-1β in the cell medium of both cell lines treated with D-Kyn (Fig. 3C,D). RNA-seq analysis of treated THP-1-derived M1 macrophages suggested closer similarity between M1 and L-Kyn-treated cells (Fig. 3E). In contrast, distinct differences were observed between M1 and D-Kyn-treated cells, as confirmed by clustering analysis (Fig. 3F). Through analysis of differentially expressed genes, we observed a significant down-regulation of PHGDH, which was established as a vital gene involved in macrophage differentiation (Wang et al, 2024), in the D-Kyn treated-macrophages, comparing to its isomer L-Kyn-treated group (Fig. 3G,H). A lower level of PHGDH was verified in mRNA levels in the D-Kyn-treated macrophages in vitro (Fig. 3I). These findings suggest D-Kyn exerts a stronger immunomodulatory effect on M1 macrophages compared to L-Kyn, potentially through PHGDH regulation.

## D-Kyn inhibits M1 macrophages inflammatory secretion via the PHGDH/NLRP3/Caspase-1 pathway

As documented by the previous study (Wang et al, 2024), we assessed the TLR4 expression and regulation in our macrophage cell line. We observed a lower expression of TLR4 expression, consistent with a lower expression of PHGDH (Fig. 4A). Consistently, a similar differential expression of PHGDH was observed in the Mac-1 and Mac-2 (Fig. 4B), further validating PHGDH as a potential macrophage differentiation factor in the context of AR. D-Kyn was validated to regulate the H3K27ac levels (Fig. 4C), as previously reported (Wang et al, 2024). In the D-Kyn treated macrophage group, we observed a prominent down-regulation of NLRP3 and cleaved Caspase-1 (p10 and p20) expression, comparing to its isomer L-Kyn (Fig. 4D). Moreover, the Caspase-1 activity in these groups displayed a similar pattern. To further demonstrate whether PHGDH functions as the upstream regulator of cleaved Caspase-1 (p10 and p20), we knocked down PHGDH expression using siRNA, which resulted in silenced expression of cleaved Caspase-1. The down-regulation of PHGDH resulted in a significant reduction in pNA concentration, reflecting lower Caspase-1 activity (Fig. 4E). The loss of PHGDH led to a lower mRNA level of IL-1β (Fig. 4F) and more distinguishable reduction in IL-1β secretion (Fig. 4G). To study the differential regulation of D- and L-Kyn in suppressing inflammatory secretion, we cultured THP-1 and primary BMDM cells, which were further treated with MCC950 for complete suppression of the NLRP3 pathway. In both cell types, the suppression of IL-1β and IL-18 was more effective in L-Kyn-treated groups, with both cytokines reduced to normal levels following MCC950 treatment (Fig. 4H). We induced NLRP3 overexpression via transfection of NLRP3 plasmid, which was verified by WB (Fig. 4I). Consistently, overexpression of NLRP3 restored the hampered secretion of both IL-1β and IL-18 (Fig. 4J). Collectively, these findings suggested that comparing to its chiral isomer L-Kyn, D-Kyn inhibits M1 macrophages' inflammatory secretion via the PHGDH/NLRP3/Caspase-1 pathway.

## D-Kyn suppresses PHGDH expression through ERK1/2/Fos pathway

To study the differential expression of PHGDH mRNA, we screened transcription factors displaying a similar expression difference. Through analyzing the differentially expressed transcription factors, we observed a higher level of FOS in Mac-1 (Fig. 5A,B). We observed a distinct reduction in the FOS protein expression under different concentrations of D-Kyn treatment (Fig. 5C). Moreover, its upstream activator ERK1/2, displayed a similar pattern in the phosphorylation levels. By utilizing the selumetinib (0.1 μM), a selective ERK inhibitor, significantly reduced the ERK phosphorylation in vitro, leading to a decrease

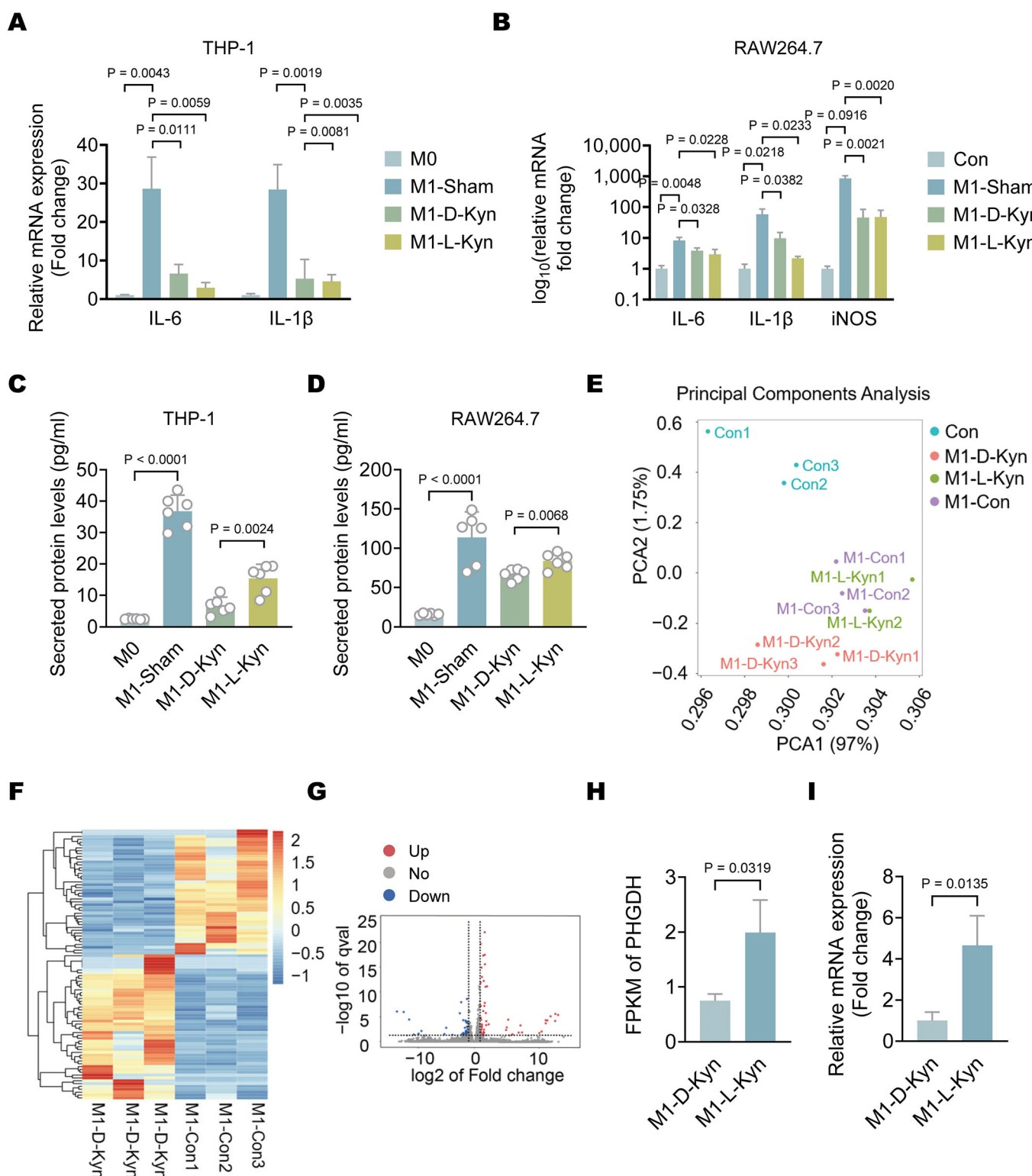

in FOS protein expression (Fig. 5D). The application of the ERK inhibitor reduced PHGDH mRNA levels (Fig. 5E). We further conducted Chip-PCR analysis to study whether Fos binds to the promoter region of PHGDH in vitro. The M1 induction significantly enhanced the binding between Fos and PHGDH promoter, which was partially hampered by the D-Kyn treatment (Fig. 5F). Through WB analysis, direct comparative analysis of D-Kyn and L-Kyn on ERK1/2 phosphorylation and FOS expression was applied. In THP-1 cells, D-Kyn displayed significantly higher potency in suppressing phosphorylation of ERK1/2 (Fig. 5G). A

**Figure 3.   D-Kyn displays higher potency in suppressing the inflammatory secretion of M1 macrophages than L-Kyn.**

(A) Treatment of D-Kyn and L-Kyn (0.1 mM) triggered a significant down-regulation of IL-1β and IL-6 mRNA levels in the THP-1 cell line ($n = 3$). (B) Treatment of D-Kyn and L-Kyn (0.1 mM) triggered a significant down-regulation of iNOS, IL-1β, and IL-6 mRNA levels in the RAW264.7 cell line ($n = 3$). (C) Elisa analysis assessing the IL-1β secretion in the THP-1 cell medium under the indicated treatment ($n = 6$). (D) Elisa analysis assessing the IL-1β secretion in the RAW264.7 cell medium under the indicated treatment ($n = 6$). (E) PCA analysis of the RNA sequencing results of different groups of THP-1 induced macrophages under indicated treatments. (F) The heatmap depicts clustering analysis results of the M1-D-Kyn and M1-Sham groups, revealing a prominent difference in gene expression patterns. (G) The volcano plot displaying the differentially expressed genes between the M1-D-Kyn and M1-L-Kyn groups. (H) FPKM derived from RNA-sequencing analysis of PHGDH assessed in M1-D-Kyn and M1-L-Kyn groups. (I) Treatment of D-Kyn triggered a significant down-regulation of PHGDH mRNA levels in the THP-1 cell line ($n = 3$). Data are presented as mean ± standard deviation, $*p < 0.05$; $**p < 0.01$; $***p < 0.001$; $****p < 0.0001$, t-test was used between the two groups; for comparisons involving three or more groups, one-way ANOVA and appropriate post-hoc multiple comparison tests were applied. Source data are available online for this figure.

similar suppressive pattern was observed in BMDMs (Fig. 5H). Collectively, these findings demonstrate that D-Kyn suppressed PHGDH expression through ERK1/2/Fos pathway.

## D-Kyn enhances allograft survival in murine skin transplantation by reducing M1 and CD8 T cell infiltration

Murine full-thickness skin transplantation, although limited in its application to humans, is a well-established in vivo model widely employed for studying alloimmune response and graft rejection (Cheng et al, 2017). We utilized this model to assess whether elevated levels of D-Kyn, as observed in patients with stable renal function, exerted any immunosuppressive effects. The injection of D-Kyn significantly postponed the progression of acute rejection, as evidenced by reduced graft shrinkage and inflammatory exudation (Fig. 6A), indicating a potential immunosuppressive influence for elevated D-Kyn levels in AR patients. H&E staining on day 14 post-transplantation revealed severe loss of cuticle, sebaceous gland, and hair follicle in the allograft experiencing AR. However, treatment with D-Kyn led to a distinguishable reduction in infiltrating immune cells and exudation (Fig. 6B). The overall survival of skin grafts indicated that D-Kyn treatment substantially ameliorated graft rejection (Fig. 6C). Immunofluorescence (IF) analysis confirmed a higher presence of M1 macrophages in the skin allograft, which was significantly reversed by D-Kyn treatment (Fig. 6D). To further investigate whether D-Kyn affected immune activation, peripheral blood from each mouse was collected and analyzed by flow cytometry. The proportion of CD8 T cells, which had expanded significantly by day 14 post-transplantation, was reduced to normal levels following D-Kyn treatment (Fig. 6E). IF analysis targeting CD4 and CD8 T cells in skin grafts showed an increased abundance of both intra-graft CD4 and CD8 T cells (Fig. 6F), but the D-Kyn treatment significantly reduced CD8 T cell abundance. IL-23a, associated with CD8 T cell activation, was also found to be decreased in the serum following D-Kyn application (Fig. EV4A). Through RNA-seq, we observed a potential down-regulation of IL-23a in both D- and L-Kyn-treated cells (Fig. EV4B), which was verified using RT-qPCR analysis (Fig. EV4C). Moreover, we observed a decrease in the protein secretion level of IL-23a in D-Kyn-treated cells (Fig. EV4D). We applied the anti-p40 antibody to block IL-23 in vivo, leading to a decrease in the intra-graft infiltration of CD8 T cells (Fig. EV4E). Collectively, these findings suggest that D-Kyn may enhance allograft survival by regulating the CD8 T cells through the modulation of IL-23a production and suppression of M1 macrophages-mediated inflammation.

## D-Kyn ameliorated the AR progressing in murine kidney transplantation

The BLAB/C to C57BL6 kidney transplant model was induced to further validate the immunosuppressive feature of D-Kyn. Application of D-Kyn for AR treatment displayed no obvious detrimental influence on renal cells (Fig. EV4F). Overall, D-Kyn treatment ameliorated AR progressing in murine kidney transplantation through various means. To gauge the overall AR-induced damage, we firstly applied H&E analysis. A significant reduction in infiltrated immune cells was found within the cortex of transplanted kidney allografts in D-Kyn-treated mice (Fig. 7A), coupled with less severe tubular cell injury (Fig. 7B). Kidney expression of NGAL, a renal injury marker, and overall TLR4 and PHGDH were shown to be down-regulated by D-Kyn application (Fig. 7C). The spiked secretion of IL-1β and IL-18 was down-regulated under D-Kyn treatment (Fig. 7D,E). Evidenced by consistent reduced signal of both CD86 and iNOS, the abundance of inflammatory macrophages was significantly reduced in D-Kyn-treated AR kidneys (Fig. 7F). To pinpoint the TLR4/ERK1/2 phosphorylation and NLRP3/Caspase-1 expression differences in the allograft-infiltrated macrophages, we applied the F4/80 magnetic beads to purify the macrophages from whole kidney cells. In consistent with the in vitro results, we observed a significantly reduced level of TLR4, PHGDH, NLRP3, and cleaved Caspase-1 expression in sorted F4/80 positive cells of the D-Kyn treated group (Fig. 7G,H). Accordingly, through quantification, we observed a lower phosphorylation level of ERK1/2 in the D-Kyn treated group (Fig. 7I). Both CD4 and CD8 positive cells were reduced in the D-Kyn treated group, which might be associated with down-regulated macrophage-secreted IL-23a levels (Fig. 7J). To detect the IL-17 secretion levels, we applied both ELISA and WB analysis. D-Kyn treatment down-regulated IL-17 transcription in the whole AR kidney (Fig. 7K), which was similar in the protein level (Fig. 7L). Based on these findings, D-Kyn significantly ameliorated acute rejection progression in murine kidney transplants by suppressing key inflammatory pathways, particularly macrophage activation via the TLR4/ERK1/2/NLRP3 inflammasome axis. Moreover, D-Kyn exerted a comprehensive immune suppressive effect on T cells.

## D-Kynurenine demonstrates superior immunosuppressive potency over L-Isomer by targeting macrophage NLRP3 inflammasome and IL-23a-mediated T cell responses

To compare the differential immune suppressive potency between D-Kyn and L-Kyn, we applied the same dosage of L-Kyn to

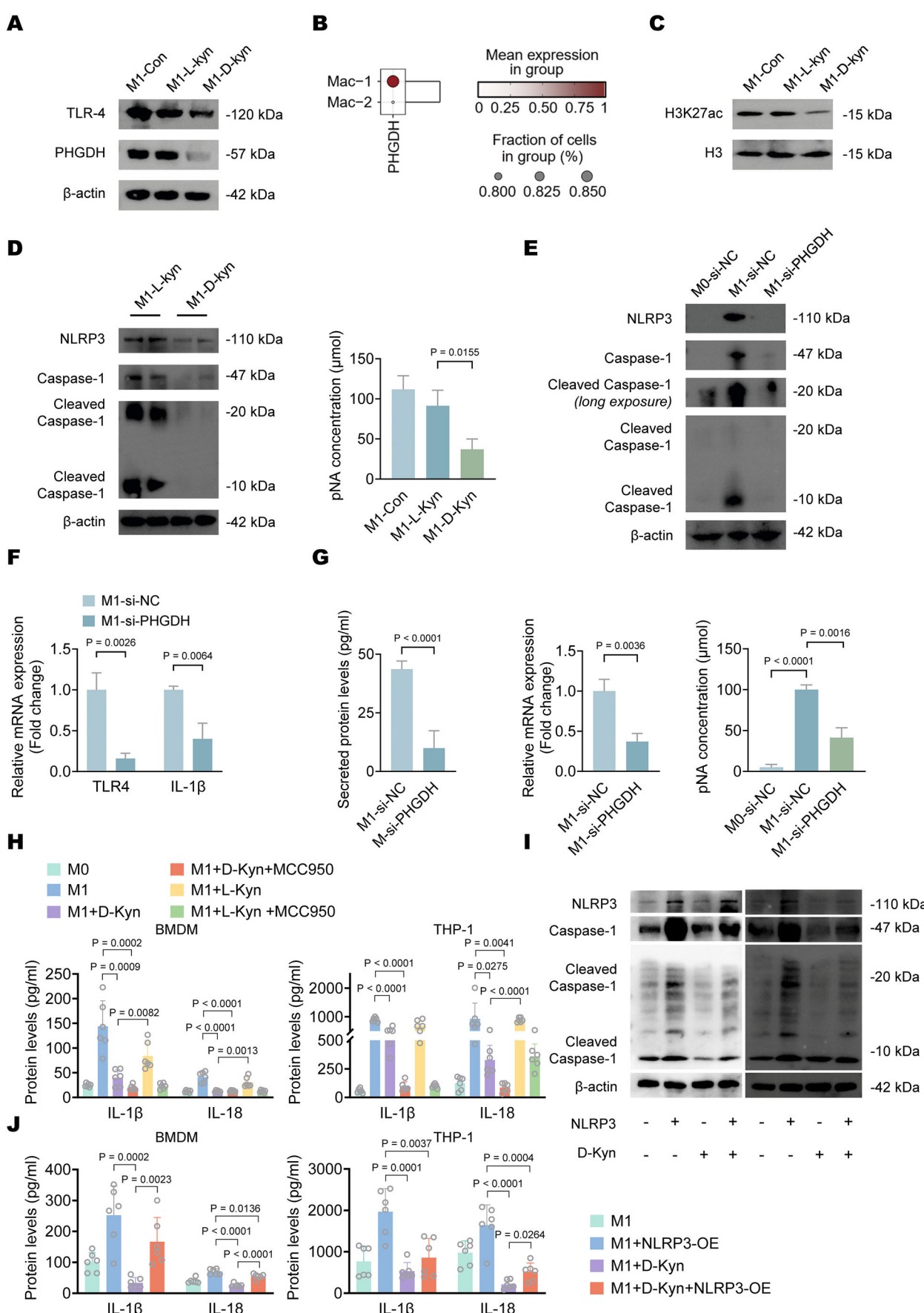

**Figure 4.  Differential effects of D-Kyn and L-Kyn on M1 macrophage activity through regulating the PHGDH/TLR4 axis.**

(**A**) WB analysis assessing differential expression of TLR4 and PHGDH expression levels in THP-1-induced macrophages. (**B**) ScRNA-sequencing results of PHGDH in the different macrophage subclusters in renal transplantation patients with rejection. (**C**) H3K27 acetylation expression of THP-1-induced M1 macrophages under indicated treatments (500 μM), using H3 as internal control. (**D**) The WB analysis assessing the differential expression of NLRP3, Caspase-1, and its cleavage product (p10 and p20) in the THP-1 cells treated with D-Kyn or L-Kyn, and the corresponding Caspase-1 activity analysis results of THP-1 cells, in which pNA concentration reflecting Caspase-1 activity ($n = 6$). (**E**) The WB analysis assessing the differential expression of NLRP3, Caspase-1 and its cleavage product (p10 and p20) in the THP-1 cells, M1-induced THP-1 cells, and M1 with PHGDH knock down. The RT-qPCR validated the partial loss of PHGDH ($n = 3$), and the corresponding Caspase-1 activity analysis results of THP-1 cells ($n = 3$). (**F**) Knockdown of PHGDH expression using siRNA resulted in a lower mRNA level of IL-1β and TLR4 ($n = 3$). (**G**) Elisa displaying a more distinguishable reduction in IL-1β secretion level ($n = 6$). (**H**) Elisa analysis of IL-1β and IL-18 secretion levels in culture supernatants of24 THP-1 cells and primary BMDMs treated with D-Kyn or L-Kyn, with or without the NLRP3 inhibitor MCC950 ($n = 6$). (**I**) Western blot analysis verifying NLRP3 and Caspase-1 activation under D-Kyn treatment and NLRP3 overexpression in THP-1 and BMDM cells. (**J**) Elisa analysis of IL-1β and IL-18 secretion in THP-1 and BMDM cells with NLRP3 overexpression, treated with D-Kyn ($n = 6$). Data are presented as mean ± standard deviation, *$p < 0.05$; **$p < 0.01$; ***$p < 0.001$; ****$p < 0.0001$, t-test was used between the two groups; for comparisons involving three or more groups, one-way ANOVA and appropriate post-hoc multiple comparison tests were applied. Source data are available online for this figure.

transplanted C57 mice. Tubular injury and overall lymphocytes infiltration remained distinct in the L-Kyn-treated AR groups (Fig. 7A), which was further supported by elevated NGAL levels (Fig. EV4G). Cleaved-Caspase-1 was partially down-regulated. However, the overall reduction of TLR4, PHGDH, and total Caspase-1 expression was insignificant (Fig. 8A). Direct comparison between D- and L-Kyn-treated groups suggested D-Kyn exerted a more significant down-regulation in the NLRP3/Caspase-1 expression (Fig. 8B), which was similar in the ERK1/2 phosphorylation and Fos expression level (Fig. 8C). To verify that the T cell immunosuppression was mediated by infiltrated macrophages, we applied CL to eliminate macrophages. Overall macrophage clearance reduced tubular injury, lymphocytes infiltration, CD8 T cell proliferation, and GZMB activation (Figs. 7A,B and 8E,G,H). Moreover, increased expression of TLR4/PHGDH and NLRP3/Caspase-1 activation was prevented by CL treatment (Fig. 8E). CL application diminished both F4/80 and inflammatory markers, CD86 and iNOS, signal intensity in AR kidneys (Fig. 7F). As regards T cell infiltration, CD8 T cells were largely reduced, while CD4 T cell abundance remained unaffected (Fig. 7J), suggesting a possible association between macrophage infiltration and CD8 T activation. Accordingly, CL treatment reduced Ki67[+] CD8 T cells, which were less prominent in the D-Kyn-treated group (Fig. 8E). Given the observation that IL-23a was derived from activated macrophages, we applied anti-IL-23a to verify the immunosuppressive effects (Fig. 8F). Blockage of IL-23a reduced overall AR injury, both CD4 and CD8 abundance, and reversed GZMB expression in the infiltrated CD8 T cells (Fig. 8G,H). Based on these in vivo evidences, D-Kyn was more effective than L-Kyn in suppressing the immune response by targeting the NLRP3/Caspase-1 pathway. This immunosuppressive effect was linked to infiltrated macrophages, as their removal reduced injury and T-cell infiltration, potentially by suppressing macrophage-derived signal IL-23a.

## Discussion

Currently, there is a growing recognition of the significant roles that D-AAs play in various physiological processes, including immune regulation. While electrochemical and chromatographic methods for enantio-analysis have been developed, the limited throughput and time efficiency reduce the translational value (Creamer et al, 2017; Kalíková et al, 2014; Shen et al, 2020). In this study, we employed a novel 17β-EBC based IM-MS approach,

which enabled efficient and sensitive separation of chiral AAs with a resolution down to 0.5% D-epimer (Li et al, 2021). This high-throughput technology demonstrated excellent enantiomeric reactivity, allowing for the rapid separation of 20 chiral AAs in a single run of approximately 2 s. With its capacity to detect the D-/L-AA ratios with such sensitivity, this technology has potential for discovering chiral disease biomarkers.

Among various D-AAs, D-Ser and D-Asp have been extensively studied for their impact on neurotransmission and immune regulation(Oliet and Mothet, 2009; Sasabe and Suzuki, 2018; Sasaki et al, 2016; Yovanno et al, 2022). The presence of certain D-AAs in the human body has been linked to multiple diseases, such as schizophrenia, atherosclerosis, and cancer (Fujii et al, 2018; Guercio and Panizzutti, 2018; Han et al, 2018; He et al, 2017; Zhang et al, 2017). However, the role of D-AAs in renal function dysregulation has yet to be fully elucidated. In the context of chronic kidney disease (CKD), serum D-Ser levels strongly correlate with the glomerular filtration ratio, serving as an essential indicator of kidney function (Hesaka et al, 2019). In our study, we identified significant differences in the stereospecific ratio of Trp and its metabolite Kyn between the AR and stable groups. A higher percentage of D-Trp in the AR group suggests its involvement in immune regulation. Previous studies have shown that D-Trp modulates immune responses by reducing Th2 cells and inducing regulatory T cells, with effects on cytokine secretion and immune cell chemotaxis (Irukayama-Tomobe et al, 2009; Kepert et al, 2017). Additionally, D-Trp has been shown to inhibit bacterial growth and biofilm formation, further underscoring its versatility in immune regulation (Cava et al, 2011; Kolodkin-Gal et al, 2010; Seki et al, 2022). Despite these findings, our study did not detect significant differences in dietary intake between AR and stable patients, suggesting endogenous regulation of D-Trp levels.

While decreased plasma Trp and elevated Kyn levels have been observed in CKD, limited evidence exists regarding the role of Kyn in transplantation (Pawlak et al, 2009). Studies have shown that indoleamine 2,3-dioxygenase (IDO), an enzyme responsible for Trp metabolism is upregulated in CKD and correlates with disease severity (Schefold et al, 2009). Although dialysis reduces Trp metabolites, the Kyn pathway remains disrupted (Mor et al, 2020). In renal transplantation, elevated Kyn/Trp ratio have been associated with incidence of graft rejection (Brandacher et al, 2007). Studies by Brandacher et al and Vavrincova-Yaghi et al indicate that monitoring Kyn/Trp ratios may serve as a useful marker for long-term graft function (Brandacher et al, 2007). These findings suggest potential implication of D-Trp and Kyn in AR,

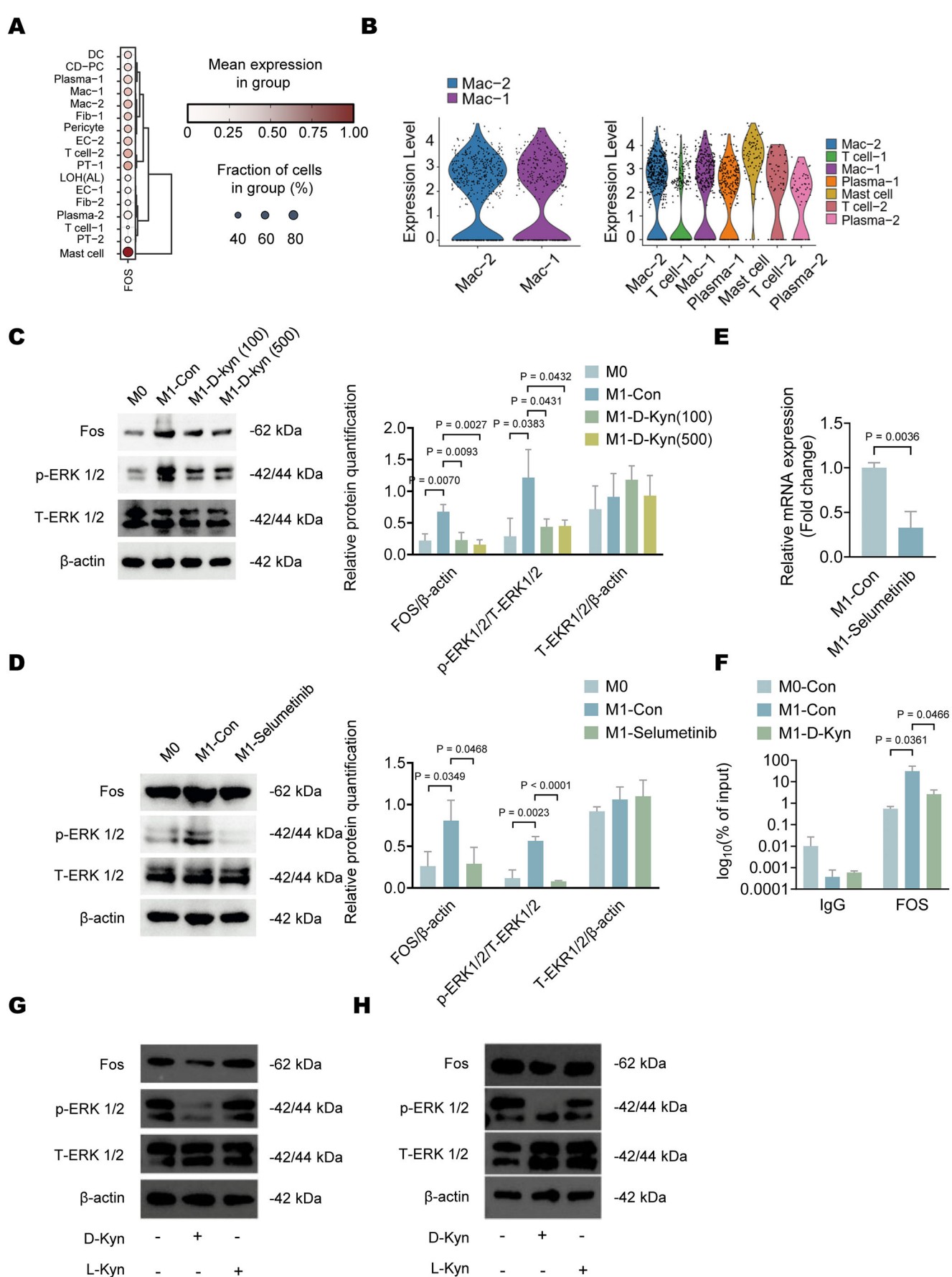

◄ **Figure 5.  D-Kyn suppresses PHGDH expression through the Fos/ERK1/2 pathway.**

(A, B) The transcription factor Fos, identified as highly expressed in macrophages and mast cells within AR kidneys, shown in the form of a dot plot (A) and violin plot (B). (C) M1-associated activation of the ERK1/2 pathway and FOS expression were inhibited by D-Kyn treatment at the indicated concentrations (0.1 mM and 0.5 mM). (D) Treatment with the selective ERK inhibitor selumetinib (0.1 μM) significantly attenuated ERK phosphorylation, leading to reduced FOS protein expression. (E) Treatment with the selective ERK inhibitor selumetinib (0.1 μM) significantly attenuated PHGDH mRNA expression ($n = 3$). (F) Chip-PCR analysis highlighting an elevated enrichment rate of FOS binding to the PHGDH promoter region, which was reduced by D-Kyn treatment ($n = 4$). (G, H) WB analysis targeting ERK1/2 activation and Fos expression in THP-1 cells (G) and BMDM cells (H). Data are presented as mean ± standard deviation, *$p < 0.05$; **$p < 0.01$; ***$p < 0.001$; t-test was used between the two groups; for comparisons involving three or more groups, one-way ANOVA and appropriate post-hoc multiple comparison tests were applied. Source data are available online for this figure.

though further research is needed to fully delineate their immune-regulatory mechanisms. Our study provides new insight into the role of D-Kyn in transplantation, particularly in its ability to mediate immunosuppression in M1 macrophages. We demonstrated that D-Kyn more effectively inhibits the secretion of inflammatory cytokines through hampering PHGDH/TLR4/Caspase-1 than L-Kyn, broadening its influence on the immune environment. A key finding was the suppression of inflammatory cytokine production by D-Kyn through down-regulating both mRNA transcription and cleavage processing mediated by Caspase-1. We observed a non-significant difference in the mRNA levels of IL-1β in the D- and L-Kyn treated macrophages, which was inconsistent with a prominent difference in the secreted protein levels in the cell medium. These results prompted us to study the regulation of IL-1β protein expression and secretion. IL-1β was a critical pro-inflammatory factor, the secretion of which was pivotally regulated by Caspase-1 cleavage (Duez and Pourcet, 2021), which was a well-established downstream factor of TLR4 (Chai et al, 2023; Liu et al, 2020). We observed the expression of active Caspase-1 was largely suppressed by D-Kyn treatment, which was mimicked by knocking down of PHGDH. These findings supported that in addition to directly promoting IL-1β mRNA expression, PHGDH facilitated IL-1β secretion via promoting Caspase-1 activation. Collectively, our study emphasized the potential of D-Kyn as a therapeutic agent, offering more efficient modulation of the immune environment than L-Kyn.

IL-23a, a versatile pro-inflammatory cytokine produced by macrophages, plays a pivotal role in regulating immune response. Its dysregulation has been linked to the exacerbation of chronic immune-mediated inflammation (Schinocca et al, 2021), through regulating CD4 and CD8 T cell activation (Ball et al, 2021). In our study, we observed D-Kyn treatment lowered IL-23a expression, which was found to be associated with intra-graft abundance of CD8 T cells, dampening the overall immune response and potentially contributing to improved graft survival. It is well-established that IL-23, comprised of IL-23a and IL-12β, is critically implicated in autoimmune disorders by prompting the differentiation of CD4 T cells into IL-17-producing Th17 lineage via STAT3 phosphorylation (Littman and Rudensky, 2010). In our mouse skin transplant AR model, D-Kyn treatment led to a reduction in CD8 T cells, corelating with lower IL-23a level, and extended survival of the allograft. This is consistent with findings in other inflammatory conditions, such as psoriasis and graft-versus-host disease, where increased IL-23 promotes CD8 T cells (Ball et al, 2021; Mehta et al, 2021). While a down-regulated Ki67+ CD8 cell proportion was found in renal transplantation model, we observed a more significant reduction in the CD4 signal abundance than CD8,

indicating immunologic differences between the two transplant models. However, given the fact that CL treatment significantly reduced secreted level of IL-23a, we proposed that D-Kyn hampered T cell activation through suppressing macrophage-derived IL-23a in acute rejection.

In the context of kidney allograft rejection, CD8 T cells are well-established mediators. The intensity of AR was shown to be associated with ratio of CD8 T cells to regulatory T cells (Mai et al, 2023). CD8 T cells can independently recognize allo-antigens and initiate allografts rejection without CD4 T cell assistance (Halamay et al, 2002). Our study demonstrates that application of D-Kyn significantly ameliorated progression of AR, reflected by various approaches. The NGAL and H&E analysis suggested a reduced tubular injury within AR kidneys treated with D-Kyn. To pinpoint the effect D-Kyn imposed upon kidney-infiltrating macrophages, we applied the F4/80+ magnetic beads to purify macrophages from the AR kidney. We observed a significantly lower level of TLR4/PHGDH, NLRP3/Caspase-1 activation and ERK1/2 phosphorylation in the macrophages treated with D-Kyn, suggesting a consistent results with our in vitro validation. Of note, we observed lower levels of IL-23a and CD86+ macrophages infiltration within AR kidney, indicating a potential regulatory approach through targeting IL-23a, specifically derived from activated M1 macrophages, underscoring the therapeutic potential of D-Kyn in modulating the immune response during transplantation. We also highlight a distinct advantage of D-Kyn over its chiral isomer, L-Kyn. While both forms of Kyn are involved in immune regulation, only D-Kyn was shown to significantly suppress M1 macrophage activity by modulating the PHGDH/TLR4 axis. This pathway is critical for the metabolic reprogramming of macrophages during inflammation. The inhibitory effect of D-Kyn on the PHGDH/TLR4 axis underscores its unique immunomodulatory properties, indicating that it may be more effective in curbing macrophage-driven inflammation compare to L-Kyn in the context of transplantation (Fig. 9).

While our study provides valuable insights into the immunosuppressive role of D-Kyn in renal transplantation, certain limitations should be acknowledged. The relatively small cohort size may affect the generalizability of our findings, although it established a solid foundation for future research, particularly in larger, more diverse patient groups. Our approach focused on determining the ratio of D-AAs to total AAs, allowing us to profile 20 D-AAs for the first time in this context and revealed significant differences between stable and AR patients, which sadly failed to achieve precise quantification of each D-AA. Future studies incorporating precise quantification could enhance the clinical relevance of these findings. Additionally, although our murine model provided valuable insights, validation in human tissues is

necessary to strengthen the clinical applicability of D-Kyn in transplantation. Despite these limitations, our findings lay the groundwork for future research aimed at refining and expanding these insights to enhance their clinical impact in transplant immunology.

# Methods

### Reagents and tools table

| Reagent/Resource | Reference or Source | Identifier or Catalog Number |
| --- | --- | --- |
| **Experimental models** | | |
| C57BL/6 J | Lingzhi Bio | C57BL/6 J WT |
| Balb/C | Lingzhi Bio | Balb/C WT |
| **Recombinant DNA** | | |
| **Antibodies** | | |
| GZMB | Abcam | ab255598 |
| CD8 | Abcam | ab209975 |
| CD4 | Abcam | ab183685 |
| CD86 | CST | 19589 |
| F4/80 | CST | 70076 |
| KI-67 | CST | 9129 |
| INOS | Proteintech | 22226-1-AP |
| β-Actin | Abclonal | AC026 |
| p44/42 MAPK (Erk1/2) | CST | 4695 T |
| Phospho-p44/42 MAPK (Erk1/2) (Thr202/Tyr204) | CST | 4370 T |
| c-Fos | CST | 31254 |
| TLR4 | Proteintech | 19811-1-AP |
| H3K27ac | CST | 8173 T |
| Histone H3 | Proteintech | 17168-1-AP |
| PHGDH | Proteintech | 14719-1-AP |
| NGAL | Proteintech | 30700-1-AP |
| c-Fos (chip) | CST | 31254 |
| PE anti-mouse CD8a (FC) | Biolegend | 100707 |
| FITC anti-mouse CD3 (FC) | Biolegend | 100305 |
| The dosage was documented in additional file 2. | | |
| **Oligonucleotides and other sequence-based reagents** | | |
| All primer sequences were documented in additional file 1. | | |
| **Chemicals, Enzymes and other reagents** | | |
| D-Kynurenine | Shanghai Yuanye Bio | S25375 |
| L- Kynurenine | Shanghai Yuanye Bio | B26381 |
| Clodronate liposomes | MCE | HY-172202 |
| anti-IL-23a (anti-p40-in vivo) | Selleck | A2163 |
| PMA | MCE | HY-18739 |
| IFN-γ | MCE | HY-P7025 |
| LPS | MCE | HY-D1056 |
| anti-F4/80 MicroBeads UltraPure | Miltenyi Biotec | 130-110-443 |
| M-CSF | MCE | HY-P7085 |

| Reagent/Resource | Reference or Source | Identifier or Catalog Number |
| --- | --- | --- |
| The dosage was documented in Methods. | | |
| **Software** | | |
| R v4.5.1 | https://www.R-project.org/ | |
| Seurat v5.3.0 | https://satijalab.org/seurat/ Hao et al, 2024 | https://doi.org/10.1038/nbt.3192 |
| dplyr v1.1.4 | https://cran.r-project.org/package=dplyr Wickham et al, 2025 | https://dplyr.tidyverse.org |
| tidyverse v2.0.0 | https://www.tidyverse.org/ Wickham et al, 2025 | https://doi.org/10.21105/joss.01686 |
| monocle v2.3.6 | https://github.com/cole-trapnell-lab/monocle-release Qiu et al, 2017 | https://doi.org/10.1038/nmeth.4402 |
| CellChat v1.6.1 | https://github.com/sqjin/CellChat Jin et al, 2021 | https://doi.org/10.1038/s41467-021-21246-9 |
| GPTCelltype v1.0.1 | https://github.com/Winnie09/GPTCelltype Hou and Ji, 2024 | https://doi.org/10.1038/s41592-024-02235-4 |
| openai v0.4.1 | https://cran.r-project.org/package=openai | https://doi.org/10.32614/CRAN.package.openai |
| patchwork v1.3.1 | https://cran.r-project.org/package=patchwork Pedersen, 2025 | https://patchwork.data-imaginist.com |
| clustree v0.5.1 | https://cran.r-project.org/package=clustree Zappia and Oshlack, 2018 | https://doi.org/10.1093/gigascience/giy083 |
| RColorBrewer v1.1.3 | https://cran.r-project.org/package=RColorBrewer | https://doi.org/10.32614/CRAN.package.RColorBrewer |
| scRNAtoolVis v0.1.0 | https://github.com/junjunlab/scRNAtoolVis Jun, 2022 | https://github.com/junjunlab/scRNAtoolVis, https://junjunlab.github.io/scRNAtoolVis-manual/ |
| presto v1.0.0 | https://github.com/immunogenomics/presto | |
| viridis v0.6.5 | https://cran.r-project.org/package=viridis | https://github.com/sjmgarnier/viridis |
| ggalluvial v0.12.5 | https://cran.r-project.org/package=ggalluvial | https://github.com/corybrunson/ggalluvial |
| ggplot2 v3.5.2 | https://ggplot2.tidyverse.org/ Wickham, 2016 | https://ggplot2.tidyverse.org |
| ggrepel v0.9.6 | https://cran.r-project.org/package=ggrepel Slowikowski, 2024 | https://ggrepel.slowkow.com/, https://github.com/slowkow/ggrepel |
| **Other** | | |

### Sample collection and preparation

A total of 23 serum samples were obtained from 12 renal transplant recipients diagnosed with AR and 11 recipients with stable kidney function. The diagnosis of AR was confirmed via biopsy, with specimens evaluated by two independent, blinded pathologists. Classification criteria followed the Banff 2019 guidelines. Serum samples were collected prior to renal allograft biopsy

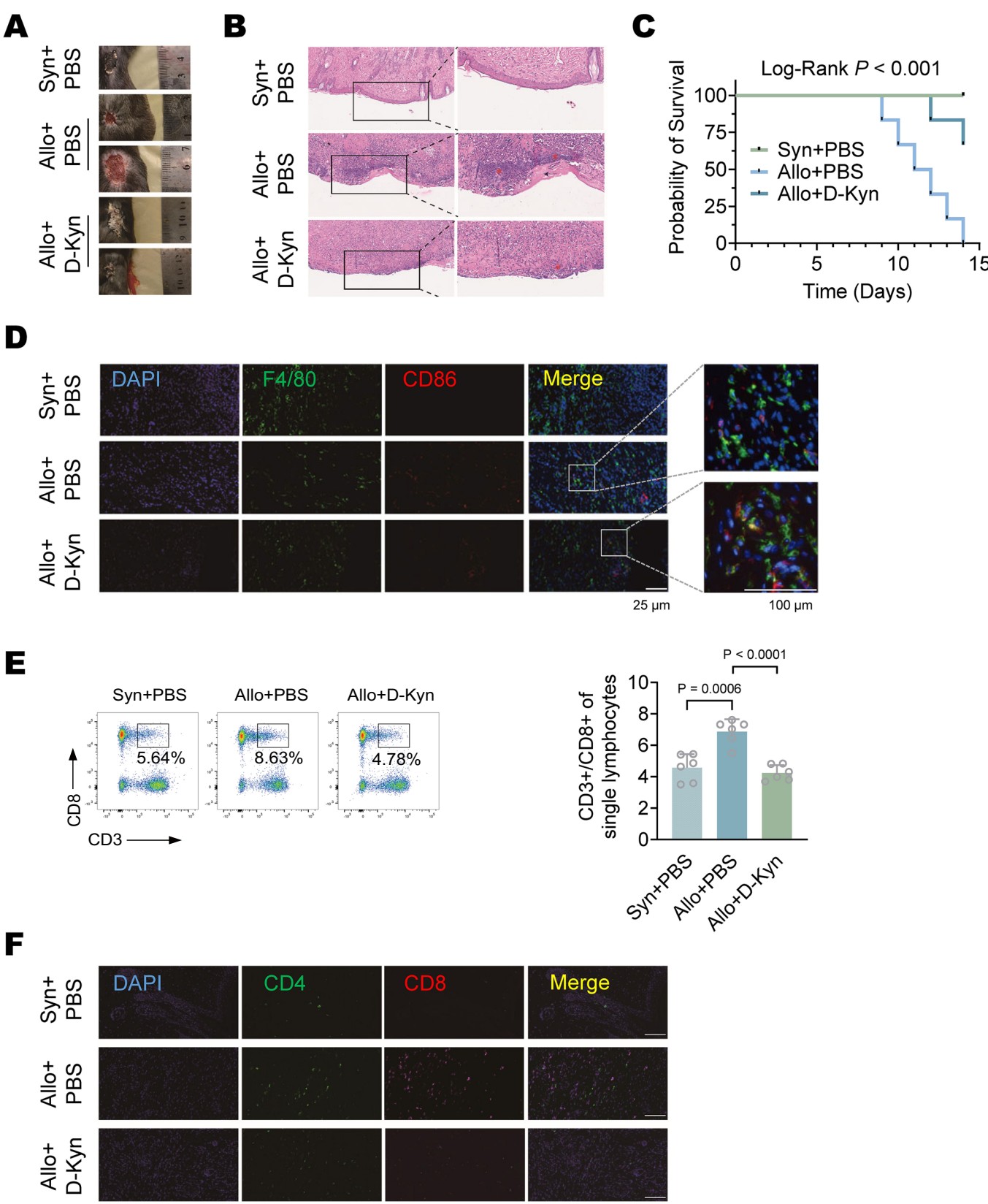

Figure 6. Immunomodulatory effects of D-Kyn on prolonging skin graft survival by inhibiting CD8 T cells and M1 macrophage.

(A) Injection of D-Kyn mildly postponed allograft loss, as evidenced by reduced levels of graft shrinkage and inflammatory exudation. (B) H&E staining of skin grafts on day 14 post-transplantation. *, clusters of infiltrated immune cells; →, severe inflammatory exudation. (C) The general KM analysis of three groups of C57BL6 mice receiving syngeneic, allogeneic graft, and allogeneic graft with D-Kyn treatment ($n = 6$). (D) IF analysis revealing D-Kyn significantly reduced the activation of M1 macrophages, labeled by positive CD86 and F4/80 staining (magnification, 400×; scale bar: 25 μm and 100 μm, respectively). (E) Flow cytometry analysis of peripheral blood collected from mice, highlighting a significantly expanded proportion of CD8 T cells on day 14 post-transplantation ($n = 6$). (F) IF analysis revealing D-Kyn significantly postponed the CD8 T cells abundance in the skin grafts (magnification: 400×, scale bar: 25 μm). Data are presented as mean ± standard deviation, *$p < 0.05$; **$p < 0.01$; ***$p < 0.001$; ****$p < 0.0001$; t-test was used between the two groups; for comparisons involving three or more groups, one-way ANOVA and appropriate post-hoc multiple comparison tests were applied. Source data are available online for this figure.

using sterile anticoagulation tubes pre-treated with EDTA (5 ml volume). After centrifugation at 3000 rpm for 8 min at 4 °C, the supernatant was collected and stored at −80 °C until further analysis. For IM-MS analysis, working solutions were prepared by diluting 1 ml of serum with 500 μl of a mixed DMF and $Na_2CO_3$-$NaHCO_3$ buffer solution (v/v ratio of 4:1) and stored at 4 °C until use.

## IM-MS analysis

IM-MS analysis was conducted as previously published (Li et al, 2021), the parameters of which were listed in Table EV1. Briefly, an ion mobility-mass spectrometer (Shimadzu, Kyoto, Japan) was used for IM-MS analysis. The U-shaped ion mobility device, with an optical length of approximately 40 mm, was installed on a modified QqQ MS (8040, Shimadzu). Drift gas flow ($N_2$) was pumped through the drift region and controlled by a digital valve, maintaining a pressure of approximately 200 Pa upstream and 100 Pa downstream. Key operating parameters included a nebulizer gas flow of 2 L/min ($N_2$), drying gas flow of 5 L/min ($N_2$), desolating line temperature of 250 °C, and heat block temperature of 400 °C. Samples were injected at 10 μL/min using a syringe pump (Harvard Apparatus, Holliston, MA, USA). All calculations were performed using the Gaussian 16 C.01 program, with structural optimization at the B3LYP level of density functional theory and Grimme's D3(BJ) dispersion correction.

## LASSO and PLS analysis

Logistic LASSO regression was carried out using the "glmnet" package (version 2.0-16). Ten-fold cross-validation was employed to select the penalty term ($\lambda$). Binomial deviance was calculated to evaluate the model's predictive performance, and two automatic $\lambda$ values were provided. The first $\lambda$ was minimized binomial deviance, and the second represented the largest $\lambda$ within 1 standard error of the minimum. In our study, we opted for the former $\lambda$ value because it imposed a less stringent regularization penalty, thereby preventing excessive covariate loss and ensuring superior potential predictive efficacy compared to the latter $\lambda$. The $\lambda$ value that minimized by the binomial deviance in our analysis was 0.18742, while the latter $\lambda$ value was 0.43297. The minimum mean square error achieved was 0.5627.

Partial least squares (PLS) analysis was conducted using the "pls" package (version 2.8-2) to evaluate the ability of the D-AAs to distinguish AR from stable patients and to construct a predictive model. In the PLS model for evaluation, the exact calculation was listed as follows:

$$Comp1 = -5.2659448873696 * Met + 4.7687392574034 * Thr + 1.68224707192298 * Phe + -14.9345567685788 * Kyn;$$

$$Comp2 = -4.11033109511724 * Met + 4.68276561315401 * Thr + 1.89830940035866 * Phe + -15.4725029608117 * Kyn;$$

$$Comp3 = -4.00202074008789 * Met + 5.02850732078142 * Thr + 2.6335257041615 * Phe + -15.3309949975128 * Kyn;$$

$$y = -13.3782967225747 * Comp1 + 14.4800121913388 * Comp2 + 6.21408217644314 * Comp3$$

## Internal model validation

The Youden index was maximized using receiver operating characteristic (ROC) curve analysis to determine the optimal cutoff value. The veracity of the model was evaluated by sensitivity, specificity, predictive values, and likelihood ratios. Decision curve analysis (DCA), a methodology employed for the computation of clinical net benefit, was used to estimate the precision of both the selected and univariate models. The optimal model was then used to construct a nomogram, which will undergo validation in future clinical applications.

## Establishment of murine skin and kidney transplantation models

We utilized a Balb/c to C57BL/6 (complete mismatch) full-thickness skin transplantation to mimic the incidence of alloimmune response and acute rejection as previously described (Cheng et al, 2017). Briefly, a skin graft (20 mm × 15 mm) from Balb/c mice was transplanted onto C57BL/6 mice. D-Kyn-treated mice were subjected to intraperitoneal injections of 40 mg/kg/d D-Kynurenine dissolved in normal saline (NS) or NS alone, administered every other day for 10 days post-transplantation.

For murine kidney transplantation models, syngeneic kidney transplantation was implemented from C57BL6 to C57BL6, while allogeneic transplantation was implemented from BALB/C to C57BL6 mice. Intraperitoneal injection of D-Kynurenine, L-Kyn (40 mg/kg per dose dissolved in normal saline, NS) or NS alone,

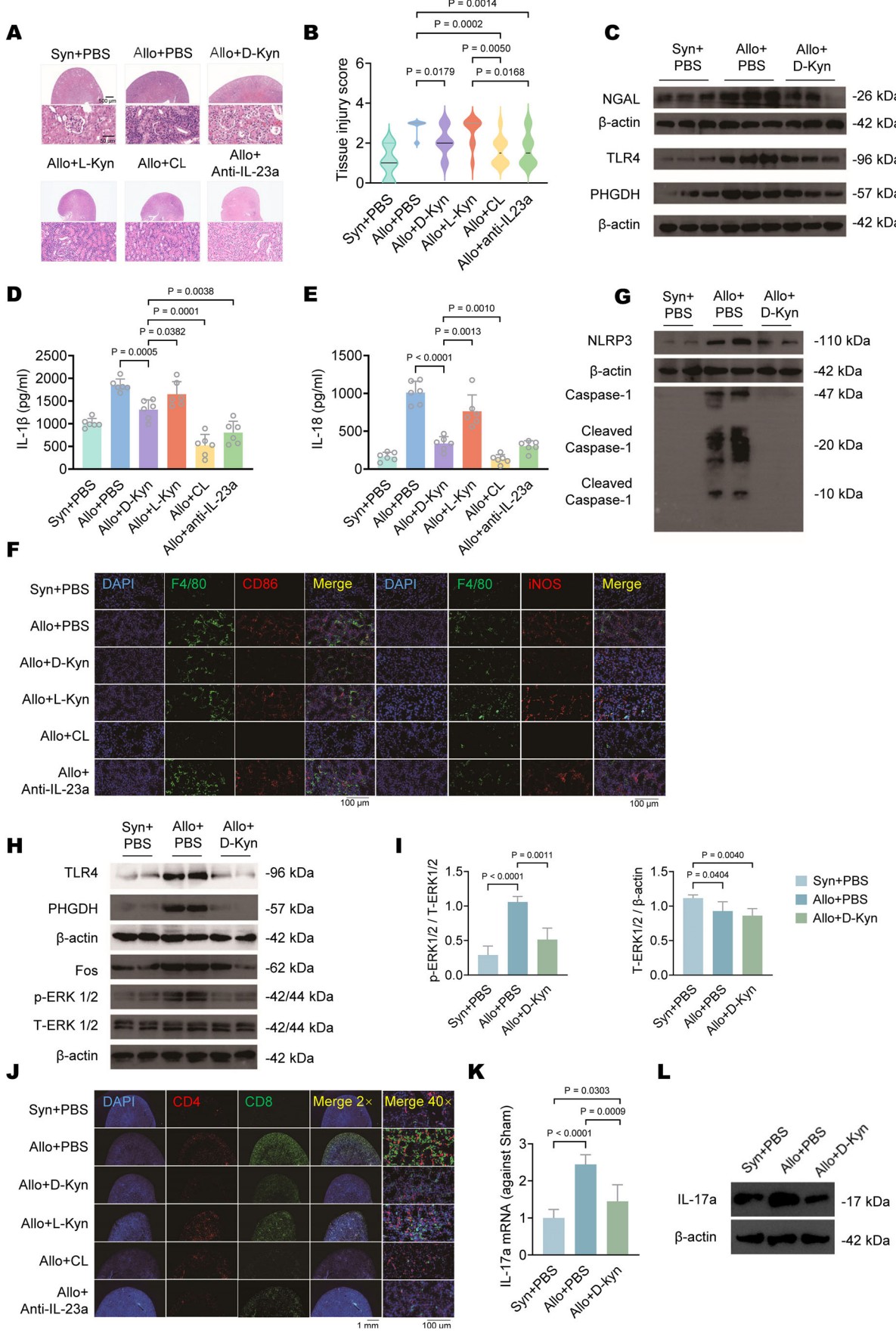

**Figure 7.   D-Kyn ameliorates acute rejection in murine kidney transplantation by suppressing macrophage inflammation and T cell infiltration.**

(A) Representative H&E-stained sections of kidney allografts 7 days post-op, showing reduced cortical immune cell infiltration in D-Kyn-treated mice. (B) Tubular injury score of transplanted kidneys in the indicated groups (10 random HP fields per group). (C) Western blot analysis showing protein levels of renal NGAL, TLR4, and PHGDH in whole kidney tissue. (D, E) Elisa analysis of IL-1β (D) and IL-18 (E) secretion ($n = 6$). (F) IF analysis targeting CD86, iNOS, and F4/80 in grafts of different groups. (G–I) Western blot analysis of TLR4 and PHGDH, NLRP3 and cleaved Caspase-1, and p-ERK1/2 and total ERK1/2 in F4/80 magnetic bead-sorted graft-infiltrating macrophages. (J) IF analysis targeting CD4 and CD8 T cell infiltration in the whole transplanted kidney tissue. (K, L) qPCR (K) and western blot (L) analysis of IL-17a levels in kidney allografts. Data are presented as mean ± standard deviation, *$p < 0.05$; **$p < 0.01$; ***$p < 0.001$; t-test was used between the two groups; for comparisons involving three or more groups, one-way ANOVA and appropriate post-hoc multiple comparison tests were applied. Source data are available online for this figure.

administered every other day on postoperative day 0 (immediately after abdominal closure), 2, 4, and 6. For anti-IL-23a application, anti-IL-23a (anti-p40-in vivo, A2163, 4.37 mg/ml, Selleck) was administered via tail vein injection (0.2 ml for the first dose on postoperative day 0, and 0.1 ml on day 2 and day 4). For clodronate liposomes (HY-172202, MCE) administration, the first dose (100 μl) was given on day 0, the second dose (100 μl) on day 2, and the third dose (50 μl) on day 4. All grafts were harvested at day 7.

For donor surgery, we cauterized the left lumbar vein and associated vasculature; dissected and transected the left ureter, followed by isolation of the infrarenal inferior vena cava and aorta, and ligation of aortic branches. After clamping the aorta distal to the right renal artery and the inferior vena cava, we transected the left renal vein and flushed the aorta, which was preserved in 4 °C NS. For recipient surgery, we exposed and isolated the abdominal aorta and inferior vena cava, secured with microvascular clamps. Openings in the aorta and inferior vena cava were made for end-to-side anastomosis with the donor renal artery and vein. For ureteral implantation, we tunneled the donor ureter through the bladder wall and trimmed the distal ureter, and inserted it into the bladder lumen, followed by suturing the bladder incision.

### Cell culture and separation

The THP-1 human monocyte and RAW264.7 murine macrophage cell lines were kindly gifted from Dr. Shi, which were cultured in RPMI-1640 and Dulbecco's modified Eagle's medium (DMEM)-high glucose medium, respectively. D-Kyn and L-Kyn were purchased from the Shanghai Yuanye Bio, which were dissolved in PBS for a final concentration of 0.1 mM. M0 differentiation was induced using PMA (100 ng/ml) for at least 12 h. M1 differentiation was induced using IFN-γ (20 ng/ml) and LPS (100 ng/ml). For primary bone marrow-derived macrophages (BMDM) isolation, cells were harvested from the femurs of WT mice. Briefly, the bone marrow cells were collected by centrifugation and then cultured for 5 days in a special medium containing macrophage colony-stimulating factor (M-CSF). Renal macrophages were isolated using anti-F4/80 MicroBeads UltraPure (130-110-443, Miltenyi Biotec), in accordance with the manufacturer's protocol. Briefly, half of the kidney was prepared as a single-cell suspension, which was sieved and re-suspended in MACS Running Buffer and separated using the MS columns.

### Quantitative real-time PCR

Following RNA extraction from the cells as previously described (Zhou et al, 2023), the RNA was reverse-transcribed into

complementary DNA (cDNA) using the PrimeScript RT Reagent Kit (Takara, Japan), adhering to standard protocols. The PCR reaction was then prepared using cDNA, specific primers, and SYBR Green Master Mix, in accordance with the manufacturer's guidelines. Amplification was performed on an ABI Prism 7500 Sequence Detection System (Applied Biosystems). The relative mRNA levels of the target gene were quantified using the ΔΔCt method, with ACTIN as a reference control. The sequences of the primers used in the reactions are detailed in the expandable files (Dataset EV1).

### Flow cytometry

To prepare for flow cytometry analysis, tissues and blood samples harvested from mouse models were finely minced and digested with collagenase to generate a single-cell suspension. Cells were subsequently stained with a fixable viability dye to distinguish live cells from dead ones, then permeabilized to allow antibody access. Subsequently, cells were incubated in the dark with fluorochrome-conjugated antibodies for 30 min to facilitate specific binding. The samples were analyzed using the BD FACSAria III Flow Cytometer, and data were processed and analyzed with FlowJo software. A detailed list of the antibodies used in this experiment is provided in expandable files.

### Immunoblotting and enzyme linked immunosorbent assay (ELISA) analysis

Proteins were extracted from cultured cells using radioimmuno-precipitation assay (RIPA) buffer. Following thermal denaturation, proteins were separated and transferred to polyvinylidene fluoride (PVDF) membranes in accordance with standard protocols. The membranes were then blocked with 5% skim milk and incubated with the appropriate primary antibody for 2.5 h at room temperature. A summary of the primary antibodies is available in Dataset EV2. The cell culture medium was subjected to Elisa analysis assessing the secretion of IL-23a, following the manufacturer's instructions.

### Chromatin immunoprecipitation-quantitative PCR (ChIP-qPCR) analysis

We utilized SimpleChIP® Enzymatic Chromatin IP Kit (Magnetic Beads) (9003, CST) in accord with protocols for Chip-PCR analysis. Briefly, approximately $10^7$ THP-1 cells for each immunoprecipitation. To cross-link proteins to DNA, 270 μl of 37% formaldehyde is added to 10 ml medium in each culture dish and incubate for 10 min, followed by the addition of glycine for further incubation.

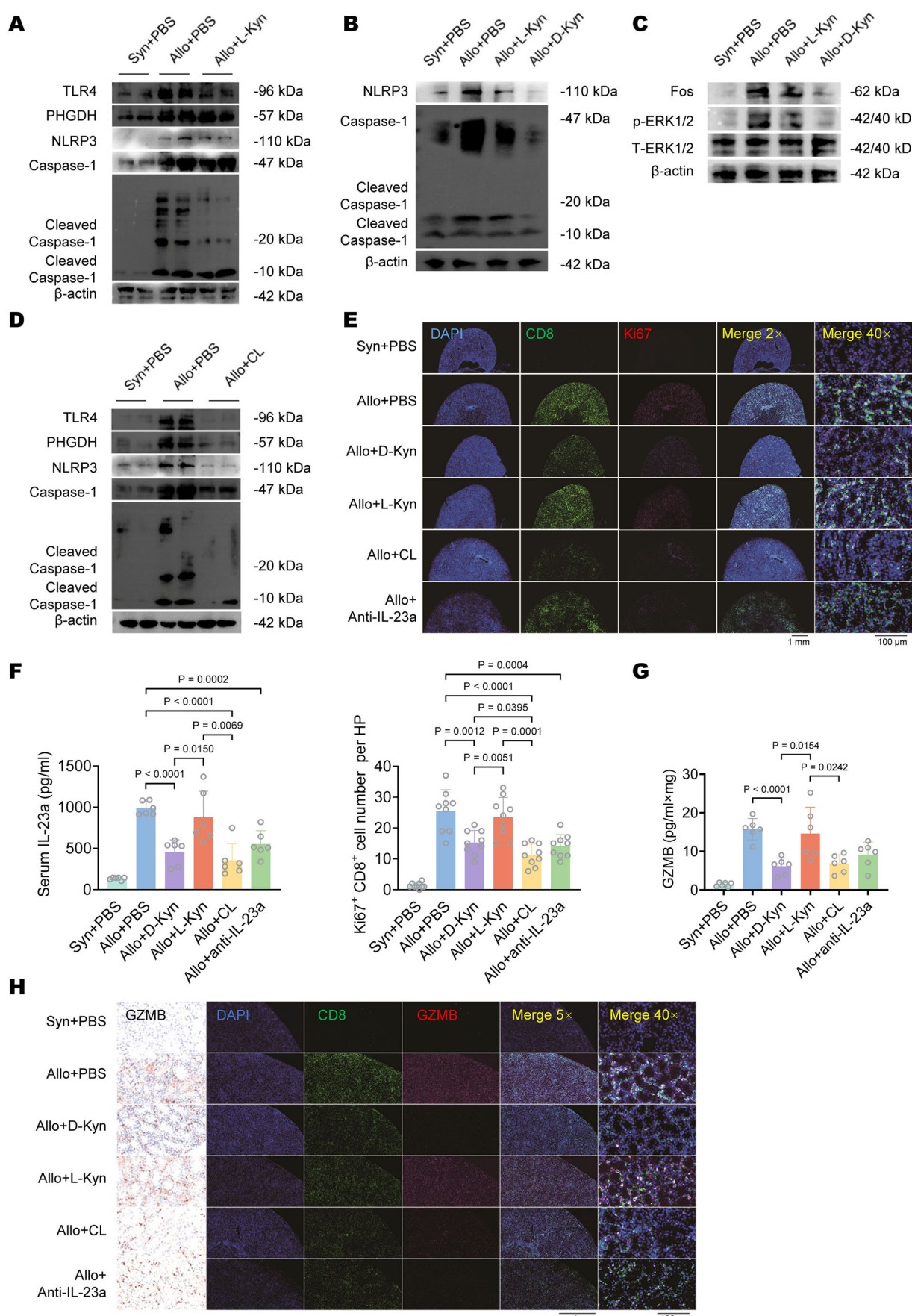

◀ **Figure 8.    D-Kynurenine demonstrates superior immunosuppressive potency over L-isomer by targeting macrophage NLRP3 inflammasome and IL-23a-mediated T cell responses.**

(**A**) Western blot illustrating TLR4/PHGDH and NLRP3/Caspase-1 pathway activation status in L-Kyn-treated allograft groups. (**B, C**) Direct comparison between D- and L-Kyn-treated allograft groups of (**B**) NLRP3 and cleaved Caspase-1, and (**C**) p-ERK1/2, total ERK1/2, and Fos. (**D**) Western blot analysis of TLR4, PHGDH, NLRP3, and cleaved Caspase-1 in allografts from CL-treated mice. (**E**) Representative image of IF analysis of Ki67 + CD8 T cells in different groups and corresponding quantification results (calculating 9 random HP fields per group). (**F**) Elisa analysis of serum IL-23a levels in different allograft groups ($n = 6$). (**G**) Elisa analysis of kidney GZMB levels in different allograft groups ($n = 6$). (**H**) IHC and IF analysis results illustrating the GZMB expression in the infiltrated CD8 cells in kidney allografts. Data are presented as mean ± standard deviation, *$p < 0.05$; **$p < 0.01$; ***$p < 0.001$; ****$p < 0.0001$; t-test was used between the two groups; for comparisons involving three or more groups, one-way ANOVA and appropriate post-hoc multiple comparison tests were applied. Source data are available online for this figure.

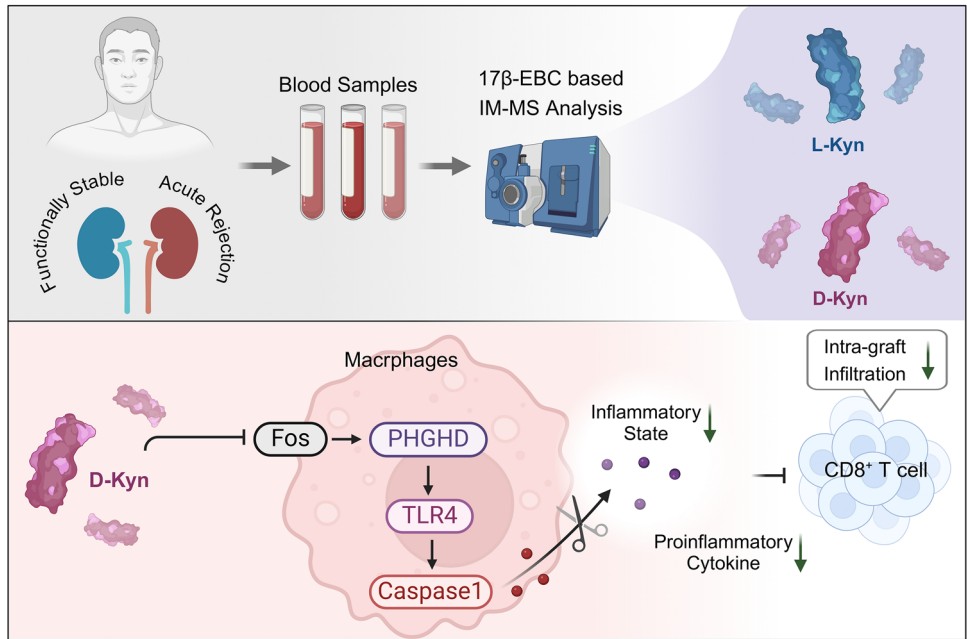

**Figure 9.    D-Kynurenine suppresses acute rejection by inhibiting the M1 macrophage PHGDH/TLR4/NLRP3 inflammasome axis and subsequent IL-23a-mediated CD8+ T cell activation.**

Schematic illustration of serum metabolite screening using chiral IM-MS, revealing that a higher proportion of D-kynurenine strongly correlated with stable graft function in kidney transplant recipients (upper). Schematic summary of the proposed mechanism by which D-Kyn ameliorates acute rejection in transplantation (Lower). In infiltrating M1 macrophages within the allograft, D-Kyn (more potently than L-Kyn), inhibit the ERK1/2/FOS signaling pathway, leading to reduced transcription of PHGDH. Downregulation of PHGDH suppresses TLR4 expression and inhibits the activation of the NLRP3 inflammasome and Caspase-1, thereby decreasing the cleavage and secretion of pro-inflammatory cytokines IL-1β and IL-18. D-Kyn treatment reduces macrophage-derived IL-23a, which attenuates the activation, proliferation, and cytotoxic function of alloreactive CD8+ T cells.

We utilized 1.5 μl micrococcal nuclease to break DNA fragments. After digesting, we stopped the reaction with EDTA, pelleted the nuclei again, and resuspended them in ChIP Buffer. The QSONIC ultrasound was utilized to break the nuclei (on:30 s and off:30 s, approximately 15 min in total) till the solution reached full transparency. 20 μl c-Fos antibody was added to each sample for ChIP.

## Caspase-1 activity analysis

We utilized the Caspase-1 activity assay kit (abs50023, Absin) for analysis. Briefly, THP-1 cells were differentiated into M0 macrophages by treatment with 100 nM PMA for 24 h, which were subsequently detached using trypsinization, pooled with the reserved medium, and centrifuged at $600 \times g$ for 5 min at 4 °C. The supernatant was carefully removed and resuspended in lysis

buffer (100 μL), incubated on ice for 15 min, and centrifuged at 12,000 rpm for 10 min at 4 °C. 50 μL lysate was incubated with Ac-YVAD-pNA at 37 °C for 60 min, which was quantified by measuring absorbance at 405 nm.

## RNA-seq analysis

Total RNA was extracted using Trizol reagent, with the resulting RNA analyzed for quantity and purity using a Bioanalyzer 2100 and RNA 6000 Nano LabChip Kit. mRNA was purified from 5 μg of total RNA through two rounds of purification using Dynabeads Oligo (dT). The purified mRNA was then fragmented into shorter pieces using divalent cations at elevated temperatures. These RNA fragments underwent reverse transcription to create cDNA with SuperScript II Reverse Transcriptase, followed by the synthesis of U-labeled second-stranded DNAs using *E. coli* DNA polymerase I

and RNase H. An A-base was added to the blunt ends to facilitate ligation with indexed adapters, which featured T-base overhangs. After ligation, size selection was performed, and the U-labeled second-stranded DNAs were treated with a heat-labile UDG enzyme and amplified via PCR. The final cDNA library had an average insert size of $300 \pm 50$ bp, and paired-end sequencing ($2 \times 150$ bp) was performed on an Illumina Novaseq 6000 following recommended protocols.

## ScRNA-seq data procurement and analysis

ScRNA-seq data were acquired from the Gene Expression Database (GSE109564), comprising samples from renal transplant patients with acute rejection (Wu et al, 2018). The datasets included in this study were publicly accessible. The "Seurat" R package was used to integrate the samples, filter, and correct the ScRNA-seq data (Chen et al, 2019). Cells with fewer than 300 or more than 4000 genes expressed, or with over 30% mitochondrial gene content, were excluded. Additionally, the DimHeatmap function in Seurat was used to examine the main sources of variation in the datasets. Clustering was performed using the FindClusters function with a resolution from 0.1 to 1 in intervals of 0.1. The "clustree" R package was used to determine that a resolution of 0.9 provided the finest and interpretable clustering.

## Cell annotation

Both manual and algorithmic annotations were performed to identify cell types more precisely. Differentially expressed genes with a log fold change >0.25 and a minimum percentage of 25% were identified as markers of cell clusters using the FindAllMarkers function in the Seurat package by the Wilcoxon rank sum test. Then, preliminary annotation results were obtained through the "GPTCelltype" R package, the ACT (Annotation of Cell Types) tool on the Cellmarker web server, alongside comparisons with the markers from the original article (Hou and Ji, 2024; Quan et al, 2023). The target genes dot plot and volcano plot for each cell population were generated using the "scRNAtoolVis" R package (Zhang, 2022).

## Construction of cell developmental trajectories

The "monocle" R package was utilized to analyze cell state in a pseudo-temporal manner (Trapnell et al, 2017). The cell dataset object was constructed by extracting the expression matrix, metadata, and cell identities from the Seurat object. Pseudo-temporal analysis results were projected into a 2-dimensional space using the "DDRTree" method. The plot_genes_in_pseudotime function was used to visualize the expression changes of IL-1β and KYNU over pseudo-time progression.

## Cell interaction analysis

Intercellular communication analyses were performed using the "CellChat" R package to further investigate the interactions between macrophages and other cells. Utilizing the receptor-ligand information in CellChatDB, potential cellular communication relationships were identified.

## Data processing

Data on gender, age, HLA mismatch, and immunosuppressant regimens are expressed as mean ± standard deviation (SD). All statistical analyses were conducted using GraphPad Prism 10 software or R software version 4.2.2 with the following packages: ggcor (correlation), glmnet, pls, rms (nomogram), dcurves (dca), tidyverse, as indicated. For comparisons between two groups, statistical significance was determined using Student's t-test, assuming normal distribution of the data. For comparisons involving three or more groups, one-way ANOVA followed by appropriate post-hoc multiple comparison tests (Tukey's or Dunnett's test) was applied. All statistical tests were two-sided, and a $P$-value less than 0.05 was considered statistically significant.

## Ethnics

Animal experiments in this study were conducted in accordance with the Guide for the Care and Use of Laboratory Animals published by the US National Institutes of Health. All mice experiments were conducted under the agreement of the Animal Experimentation Ethics Committee of Zhongshan Hospital, Fudan University (experimental license no. 2024-026). Human specimens were obtained following the approval of the Ethics Committee of Zhongshan Hospital, Fudan University, in accordance with the principles of the Declaration of Helsinki (Approval number: B2022-492). Prior to the study, written informed consent was obtained from all participants.

### The paper explained

#### Problem
Acute rejection (AR) remains a major cause of graft failure following kidney transplantation. Although dextrorotatory-amino acids (D-AAs) have been recognized as biologically active compounds, its role in mediating immunosuppression was poorly depicted.

#### Results
Kidney transplant recipients with stable graft function had significantly higher serum levels of D-Kyn compared to those experiencing AR. D-Kyn, more potently than its chiral counterpart L-Kyn, suppresses the pro-inflammatory activity of M1 macrophages through suppressing the PHGDH/TLR4/NLRP3 inflammasome axis, which in turn reduces the activation of Caspase-1 and the subsequent cleavage and secretion of inflammatory cytokines like IL-1β and IL-18. D-Kyn treatment reduced the activation and infiltration of CD8 + T cells in allografts. This effect is potentially mediated by D-Kyn's ability to inhibit macrophage-derived IL-23a. In murine models of both skin and kidney transplantation, administration of D-Kyn significantly prolonged allograft survival, attenuated macrophage-mediated inflammation, and reduced T cell infiltration and activation.

#### Impact
These findings position D-Kyn as a promising new therapeutic candidate for preventing organ transplant rejection and suggest that monitoring D-Kyn levels could serve as a prognostic biomarker for graft stability.

## Conclusion

The immunosuppressive effects of D-Kyn highlight its potential as a therapeutic agent for ameliorating acute rejection in renal transplantation by modulating macrophage-mediated inflammation through the PHGDH/NLRP3/Caspase-1 axis. Our findings provide a novel approach for AR treatment and lay the foundation for future clinical applications from the perspective of dextrorotatory amino acids.

## Data availability

The RNA-seq data has been up-loaded onto the ENA database in an open-source form, which could be accessed by accession number of PRJEB104604. All the IM-MS data was enclosed in the supplemental data for better accessibility. We declared that no exclusive material has been used in this study and all commercial products were provided with item details.

The source data of this paper are collected in the following database record: biostudies:S-SCDT-10_1038-S44321-026-00377-w.

## Peer review information

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

## Acknowledgements

We would like to thank Boyang Yu (Fudan University) for coordinating the manuscript revision. The detailed funding status was listed as follows: Fujian Provincial Health Technology Project (2020GGB058), National Natural Science Foundation of China (grants 82170765, 82370754 to CY; 82070772 to TYZ), Shanghai Municipal Key Clinical Specialty (shslczdzk05802), Shanghai Municipal Health Commission Collaborative Innovation Cluster Project

(2024CXJQ02 to TYZ), and Clinical Research Funds of Zhongshan Hospital (ZSLCYJ202335).

## Author contributions

**Yufeng Zhao**: Methodology; Writing—original draft; Project administration. **Jiaheng Wu**: Software; Formal analysis; Visualization; Methodology; Project administration. **Yuling Li**: Formal analysis; Methodology; Project administration. **Yirui Cao**: Project administration. **Tongyu Zhu**: Conceptualization; Resources; Funding acquisition. **Yinlong Guo**: Conceptualization; Formal analysis; Methodology; Writing—review and editing. **Cheng Yang**: Conceptualization; Formal analysis; Supervision; Funding acquisition; Methodology; Writing—review and editing. **Dong Zhu**: Conceptualization; Project administration; Writing—review and editing.

Source data underlying figure panels in this paper may have individual authorship assigned. Where available, figure panel/source data authorship is listed in the following database record: biostudies:S-SCDT-10_1038-S44321-026-00377-w.

## Disclosure and competing interests statement

The authors declare no competing interests.

# Expanded View Figures

**Figure EV1.  Further details of model construction.**

(**A**) The LASSO algorithm results determine the number of D-AAs for model establishment as four. (**B**) The determination of the number of components for the PLS model construction with reference to RMSEP. (**C**) The nomogram development for clear visualization of the final risk of AR incidence by different values of each D-AA.

▶

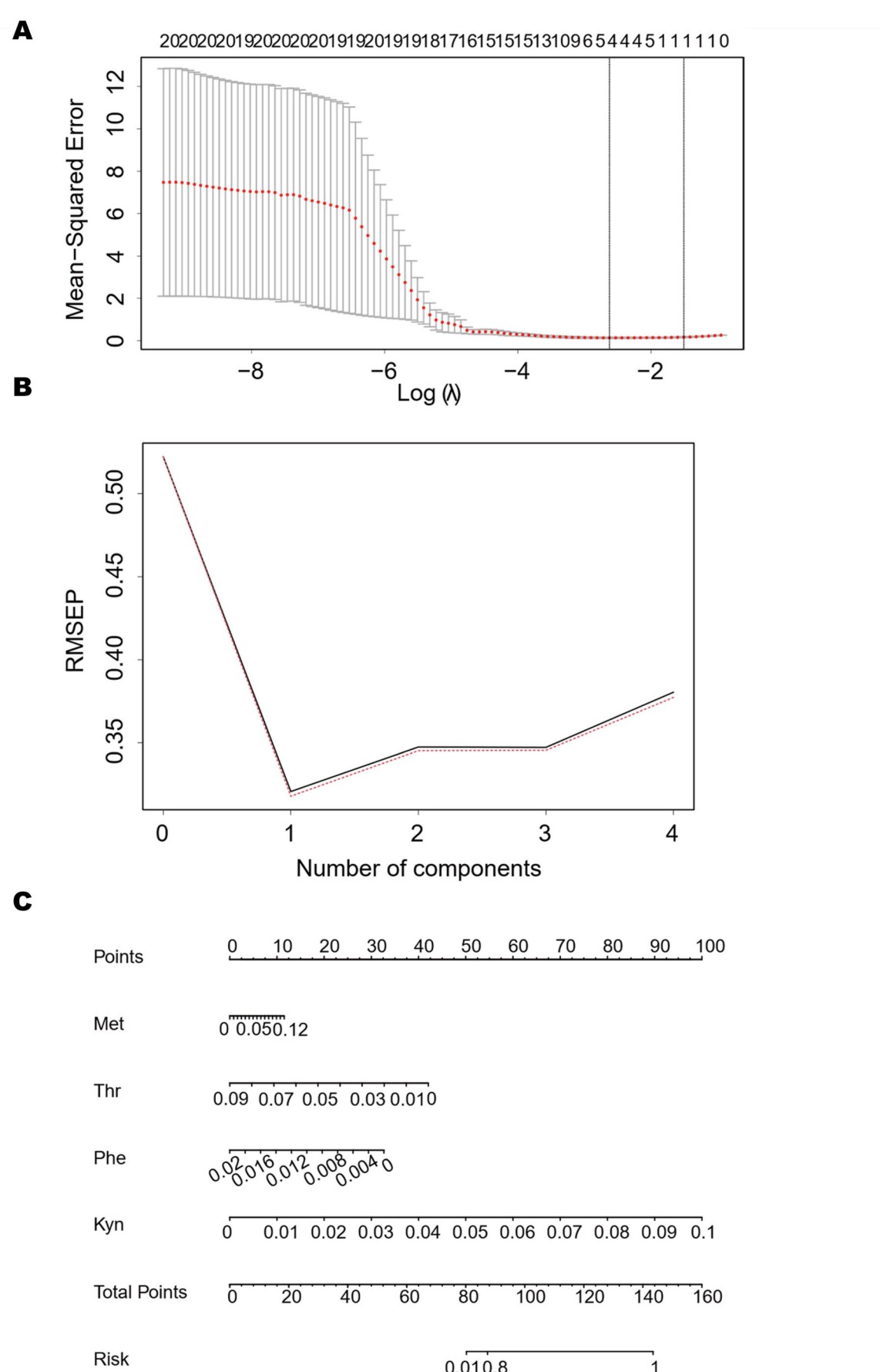

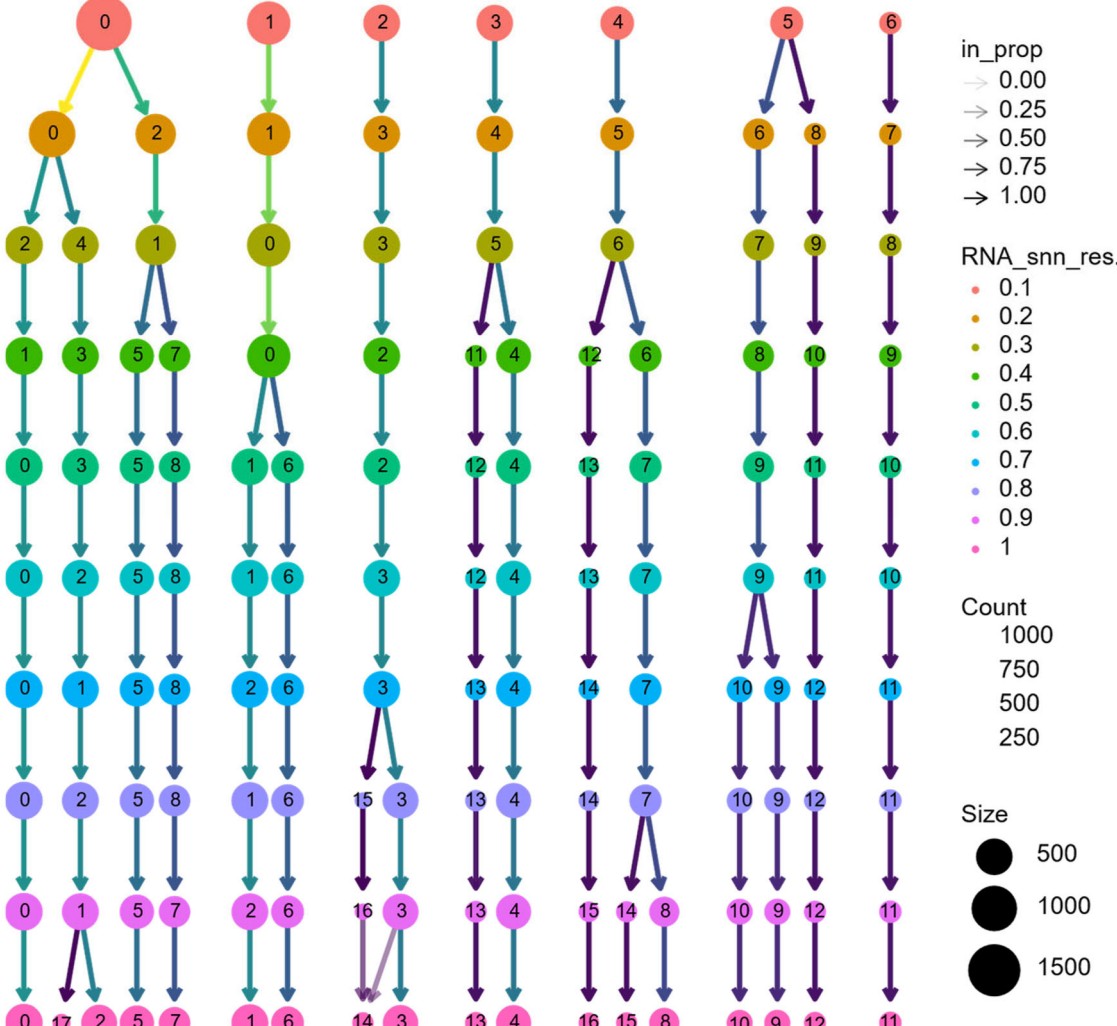

**Figure EV2.  The cell clustering strategy for scRNA-seq of the AR kidneys.**

The clustree analysis results identified a total of 16 subclusters for the whole cell population of AR kidney using the "clustree" R package, with a resolution of 0.9, providing the finest and interpretable clustering.

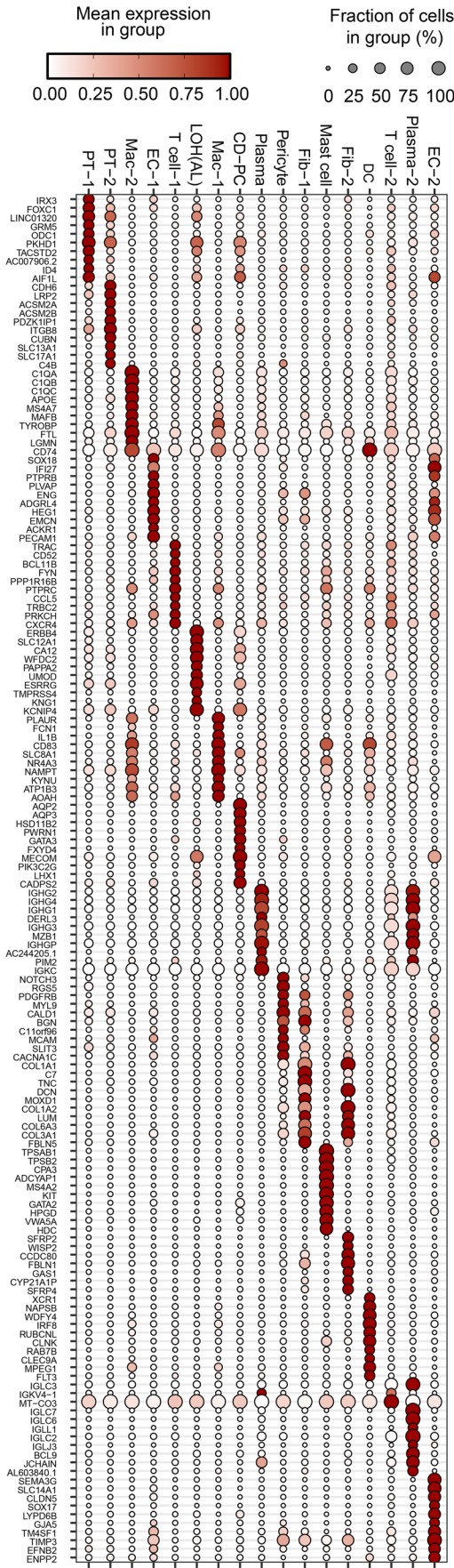

◄ **Figure EV3.  The cell annotation strategy for scRNA-seq of the AR kidneys.**

The bubble plot shows the top 10 marker genes for each cluster.

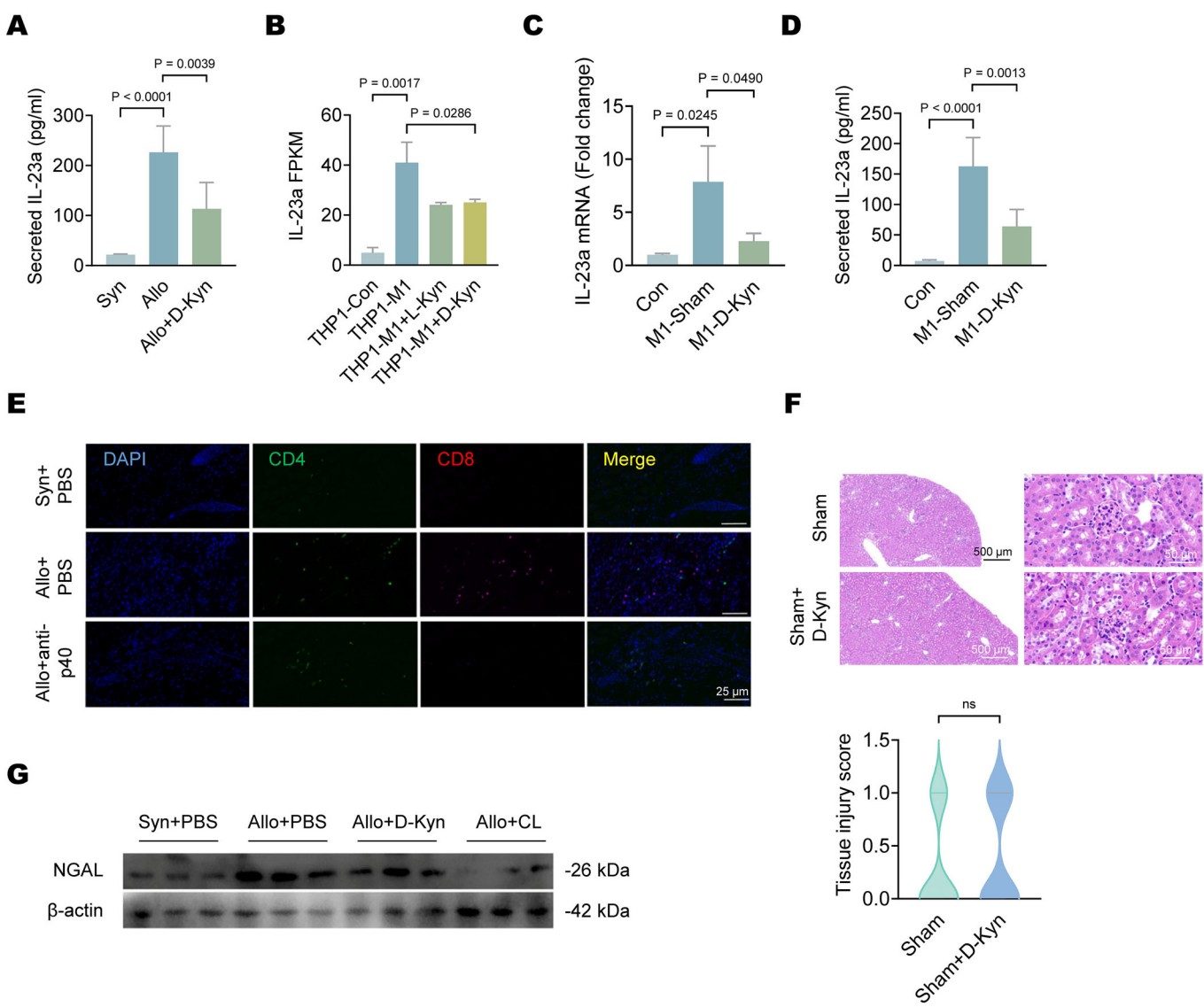

**Figure EV4. D-Kyn down-regulated CD8 T cell intra-graft infiltration, potentially via inhibiting IL-23 production.**

(A) The ELISA analysis assessing the serum IL-23a levels on day 14 post-transplantation in a skin transplant murine model ($n = 6$). (B) Comparative analysis of IL-23a expression levels in control, M1 macrophage groups treated with D-Kyn or L-Kyn, assessed by RNA-seq. (C) RT-qPCR revealed a prominent decrease in IL-23a levels under the treatment of D-Kyn (0.1 mM). (D) The significantly decreased secreted levels of IL-23a in the THP-1 cell culture medium under the treatment of D-Kyn ($n = 6$). (E) IF analysis revealing anti-p40 antibody (500 μg twice a week) significantly postponed the CD8 T cells abundance in the skin grafts (magnification: 400×, scale bar: 25 μm). (F) No significant difference between Sham mice and D-Kyn-treated mice, revealed by H&E and quantification. (G) WB analysis targeting NGAL expression in the different allograft groups. Source data are available online for this figure.

