## [Peer Review File · EMBO Molecular Medicine]

Dextrorotatory Kynurenine Suppresses Acute Rejection via Inhibiting Macrophage-Mediated Inflammation

Yufeng Zhao, Jiaheng Wu, Yuling Li, Yirui Cao, Tongyu Zhu, Yinlong Guo, Cheng Yang, and Dong Zhu

Corresponding authors: Dong Zhu (zhu.dong@zs-hospital.sh.cn) , Yinlong Guo (yiguo@sioc.ac.cn), Cheng Yang (yang.cheng1@zs-hospital.sh.cn)

Review Timeline:

Submission Date:	30th Jun 25
Editorial Decision:	25th Jul 25
Revision Received:	21st Oct 25
Editorial Decision:	26th Nov 25
Revision Received:	31st Dec 25
Accepted:	13th Jan 26

Editor: Jingyi Hou

Transaction Report:

25th Jul 2025

Dear Dr. Zu,

Thank you again for submitting your work to EMBO Molecular Medicine. We have now received feedback from the two referees who agreed to evaluate your manuscript. As you will see in the reports below, the referees find the manuscript to be of interest and novel. They raise, however, several important points, which should be convincingly addressed in a revision of this work.

I think the referees' recommendations are clear and need not be repeated here. In particular, the raised concerns regarding the mechanisms, statistics, and the comparison between D-Kyn and L-Kyn across several experiments, which need to be carefully addressed.

As you may already know, our editorial policy allows in principle a single round of major revision so it is essential to provide responses to the referees' comments that are as complete as possible. Please feel free to contact me in case you would like to discuss in further detail any of the issues raised by the referees.

Please also contact us as soon as possible if similar work is published elsewhere. If other work is published, we may not be able to extend the revision period beyond three months.

I look forward to receiving your revised manuscript.

Yours sincerely,
Jingyi

Jingyi Hou
Senior Editor
EMBO Molecular Medicine

We require:

- 1) A .docx formatted version of the manuscript text (including legends for main figures, EV figures and tables). Please make sure that the changes are highlighted to be clearly visible.
- 2) Individual production quality figure files as .eps, .tif, .jpg (one file per figure). For guidance, download the 'Figure Guide PDF': (<https://www.embopress.org/page/journal/17574684/authorguide#figureformat>).
- 3) A .docx formatted letter INCLUDING the reviewers' reports and your detailed point-by-point responses to their comments. As part of the EMBO Press transparent editorial process, the point-by-point response is part of the Review Process File (RPF), which will be published alongside your paper.
- 4) A complete author checklist, which you can download from our author guidelines (<https://www.embopress.org/page/journal/17574684/authorguide#submissionofrevisions>). Please insert information in the checklist that is also reflected in the manuscript. The completed author checklist will also be part of the RPF.
- 5) Please note that all corresponding authors are required to supply an ORCID ID for their name upon submission of a revised

manuscript.

6) It is mandatory to include a 'Data Availability' section after the Materials and Methods. Before submitting your revision, primary datasets produced in this study need to be deposited in an appropriate public database, and the accession numbers and database listed under 'Data Availability'. Please remember to provide a reviewer password if the datasets are not yet public (see <https://www.embopress.org/page/journal/17574684/authorguide#dataavailability>).

12) Author contributions: You will be asked to provide CRediT (Contributor Role Taxonomy) terms in the submission system. These replace a narrative author contribution section in the manuscript.

13) A Conflict of Interest statement should be provided in the main text.

14) Please provide a 'Synopsis' to further enhance discoverability. Synopses are displayed on the journal webpage and are freely accessible to all readers. They include a short stand first (maximum of 300 characters, including space) as well as 2-5

one-sentences bullet points that summarizes the paper. Please write the bullet points to summarize the key NEW findings. They should be designed to be complementary to the abstract - i.e. not repeat the same text. We encourage inclusion of key acronyms and quantitative information (maximum of 30 words / bullet point). Please use the passive voice. Please attach these in a separate file or send them by email, we will incorporate them accordingly.

Please also provide a visual abstract to illustrate your article as a PNG file 550 px wide x 300-600 px high.

15) All Materials and Methods need to be described in the main text using our 'Structured Methods' format. According to this format, the Methods section includes a Reagents and Tools Table (listing key reagents, experimental models, software and relevant equipment and including their sources and relevant identifiers) followed by a Methods and Protocols section describing the methods, ideally using a step-by-step protocol format. The aim is to facilitate adoption of the methodologies across labs.

Please download and fill our Reagents and Tools Table template (.docx), which you can find in our author guidelines: <https://www.embopress.org/page/journal/17574684/authorguide#structuredmethods>

When submitting your revised manuscript, please do not include the Reagents and Tools Table in the Methods section of the manuscript but upload it as a separate file choosing the file type "Reagent Table"

**** Reviewer's comments ****

Referee #1 (Comments on Novelty/Model System for Author):

The primary concern is that the number of human patients recruited for metabolomics data, as well as the number of replicates for in vitro experiments, is very low.

In addition, statistical analysis of in vitro and mouse data is not explicitly indicated.

Referee #1 (Remarks for Author):

The study by Zhao et al. investigates the role of D-Kyn in allogeneic transplantation, with a particular focus on its influence on M1 macrophage activation. The topic is both novel and interesting. However, further experimental validation is necessary to substantiate the proposed D-Kyn-mediated modulation of M1 macrophages in the prevention of acute rejection (AR).

Major Comments:

Sample Size: The number of patients enrolled in the study is notably limited. We strongly recommend increasing the sample size to enhance the statistical power and reliability of the conclusions.

Mechanistic Insights into IL-1 β Regulation: Data presented in Figure 3A-D suggest that D-Kyn affects IL-1 β secretion differently from L-Kyn, despite similar mRNA expression levels. However, it remains unclear whether this effect is mediated explicitly through the avoidance of caspase-1 cleavage, via inflammasome inhibition by D-Kyn, rather than by L-Kyn. Additional mechanistic experiments are necessary to demonstrate this differential regulation directly, such as in vitro inhibition of the inflammasome. Moreover, the inhibition of caspase-1 and inflammasome engagement by D-Kyn (but not L-Kyn) should be assessed in primary human and mouse macrophages.

MAPK and FOS Signalling: The authors should compare the effects of D-Kyn and L-Kyn on ERK1/2 phosphorylation and FOS expression (as shown in Figure 5C) to clarify their respective roles in signalling pathways.

T Cell Responses and In Vivo: Both D-Kyn and L-Kyn have the potential to modulate T cell proliferation, activation, cytokine release, and Treg induction, which are factors that can significantly impact AR. To delineate the role of M1 macrophages in the effects observed with D-Kyn in in vivo models of AR, further experiments should include: macrophage depletion strategies, or blocking monocyte or T cell recruitment to inflamed tissues. Additionally, exogenous administration of L-Kyn in vivo should be tested for comparison with D-Kyn. The study should also assess Treg differentiation and cytokine production by CD4 $^{+}$ and CD8 $^{+}$ T cells in response to D-Kyn vs L-Kyn treatment in vivo. Authors should determine the effect of D-Kyn compared with L-Kyn on cytotoxicity in CD8 $^{+}$ T cells, including granzyme and perforin expression, particularly in the context of renal transplantation, where CD8 $^{+}$ T cells predominate the infiltrate.

IL-23 and IL-17 Axis: The authors should investigate in vivo whether D-Kyn-mediated reduction in IL-23 correlates with decreased IL-17 production by T cells in mouse models of AR.

T Cell Proliferation Analysis: The flow cytometry data in Figure 6E are not sufficient to support conclusions regarding CD8 T cell proliferation. More specific proliferation markers, such as Ki-67 or BrdU, should be used. Proliferation should be analyzed in relevant tissues, such as skin, renal allografts, and lymphoid organs, rather than in peripheral blood.

Minor Comment: Additional M1 macrophage markers, such as iNOS, should be examined via immunofluorescence in tissue sections to support the findings of macrophage polarisation.

The specific statistical test used for each graph should be indicated at least in the Figure Legends.

Referee #2 (Comments on Novelty/Model System for Author):

The authors employed both kidney and skin transplantation models to investigate the therapeutic potential of D-kynurenine in acute rejection, a well-designed experimental approach that offers multiple advantages for mechanistic and translational studies.

Referee #2 (Remarks for Author):

In this manuscript, Zhao et al. present novel and exciting findings on the immunosuppressive role of D-kynurenine (D-Kyn) in transplant rejection, supported by chiral analysis and comprehensive mechanistic studies using skin and kidney transplant models. The key strengths of the study include the identification of D-Kyn as a potential immunoregulatory molecule, demonstration of D-amino acid chirality influencing transplant outcomes, and the functional validation in both skin and kidney transplant models. The authors successfully link chiral metabolic regulation to macrophage-driven immune modulation through the PHGDH/TLR4/Caspase-1 axis, and explore its therapeutic relevance.

The work is highly original and mechanistically insightful. While the manuscript is of considerable merit, a few important issues should be addressed to further enhance its clarity and logic.

1. Major concerns

1.1 The mechanistic role of D-Kyn in modulating IL-23a expression and its downstream effect on CD8⁺ T cell activation is central to the study's proposed pathway. However, while this axis is supported by in vitro findings and skin transplant experiments, it remains insufficiently validated in the kidney transplant model. Specifically, functional experiments blocking IL-23a in vivo- followed by assessment of CD8⁺ infiltration and graft damage-is necessary to fully establish the mechanistic relevance of this pathway in a clinically pertinent setting.

1.2 The skin transplant model is used extensively to elucidate immunological mechanisms in this study. Although this model offers experimental tractability, its immunological context differs from that of kidney allografts. The therapeutic efficacy of D-Kyn is demonstrated convincingly in the kidney model (Figure 7), yet most mechanistic insights-particularly regarding macrophage-T cell interactions-are derived from the skin model. Greater mechanistic resolution within the kidney transplant system would enhance the translational relevance of the findings and better align the mechanistic and therapeutic claims.

2. Minor concerns

2.1 Standardize terminology by consistently using either "D-Kyn" or "D-kynurenine" throughout. Define the abbreviation at first use.

2.2 In Figure 3A/B legends, specify which cell line (THP-1 or RAW264.7) the data corresponds to.

2.3 In Section 2.3, clarify the "y = " term in the PLS analysis. Does "y" represent AR status?

2.4 In Figures 7H/I/J, clarify whether the immunoblots were performed on sorted F4/80⁺ macrophages or whole kidney lysates, as this is inconsistent between the legend and text.

2.5 Indicate the concentration of D-Kyn/L-Kyn used in all in vitro assays (Figures 3-5, S4).

2.6 Provide the time point(s) of sample collection in the kidney transplant model shown in Figure 7.

2.7 Specify the source/vendor of D-Kyn and L-Kyn used for in vivo experiments.

2.8 Although D-Kyn shows promising therapeutic potential, the authors should briefly discuss its pharmacokinetics, stability, and possible off-target effects in vivo, especially for future clinical translation.

2.9 The novel chiral profiling method is a technical highlight of this work. Emphasizing its clinical scalability and potential for broader application could further enhance translational value.

Point-by-point responses to the reviewers' comments:

We sincerely thank both reviewers for their thoughtful comments, constructive suggestions, and positive feedback on our manuscript. We greatly appreciate the opportunity to address these valuable points, which have helped us substantially improve the quality, clarity, and mechanistic depth of our work. Detailed, point-by-point responses are provided below.

Reviewer #1:**Comment 1:**

Sample Size: The number of patients enrolled in the study is notably limited. We strongly recommend increasing the sample size to enhance the statistical power and reliability of the conclusions.

Reply 1:

We fully acknowledge the reviewer's valuable concern regarding the limited clinical sample size and sincerely appreciate this thoughtful suggestion. We agree that expanding the cohort would undoubtedly strengthen the statistical power and the generalizability of our findings.

The incidence of biopsy-proven acute rejection after renal transplantation has markedly decreased in recent years due to optimized immunosuppressive regimens, which makes the prospective recruitment of such patients inherently challenging within a short timeframe. Moreover, serum samples for chiral metabolomic profiling must be freshly collected and processed under rigorously standardized conditions to prevent post-collection metabolic drift. Pre-existing biobank samples could not be utilized because even subtle variations in storage duration or freeze-thaw cycles would compromise the accuracy of D-/L-enantiomer quantification.

Given these constraints, it was technically difficult to expand the cohort within the limited revision period. We have now clarified these constraints in the revised Discussion and explicitly framed our clinical metabolomic analysis as an *exploratory discovery phase* aimed at hypothesis generation rather than definitive biomarker validation. Importantly, the subsequent mechanistic experiments *in vitro* and *in vivo* were designed to validate the biological relevance of D-kynurenine identified in this exploratory phase, thereby strengthening the overall scientific rationale despite the modest sample size.

In addition, we have included a statement in the Limitations section acknowledging this issue and outlining our plan to extend this work through prospective, multi-center studies that will allow validation of D-kynurenine-associated metabolic signatures in larger and more diverse patient cohorts. We believe that these planned efforts will further consolidate the translational relevance of our current findings.

Comment 2:

Mechanistic Insights into IL-1 β Regulation: Data presented in Figure 3A-D suggest that D-Kyn affects IL-1 β secretion differently from L-Kyn, despite similar mRNA expression levels. However, it remains unclear whether this effect is mediated explicitly through the avoidance of caspase-1 cleavage, via inflammasome inhibition by D-Kyn, rather than by L-Kyn. Additional mechanistic experiments are necessary to demonstrate this differential regulation directly, such as *in vitro* inhibition of the inflammasome. Moreover, the inhibition of caspase-1 and inflammasome engagement by D-Kyn (but not L-Kyn) should be assessed in primary human and mouse macrophages.

Reply 2:

We sincerely thank the reviewer for this insightful and constructive comment. We agree that elucidating whether D-Kyn suppresses IL-1 β secretion through direct inhibition of the NLRP3 inflammasome and subsequent caspase-1 activation is essential for a deeper mechanistic understanding.

2.1. Inhibition of the NLRP3 inflammasome:

To clarify this mechanism, we performed additional *in vitro* experiments using the selective NLRP3 inhibitor MCC950 (2 μ M, MCE). THP-1 cells and bone marrow-derived macrophages (BMDMs) were treated with D-Kyn or L-Kyn, with or without MCC950. D-Kyn markedly reduced the secretion of IL-1 β and IL-18, whereas these cytokines returned to baseline levels upon MCC950 administration (Figure 4H). These results indicate that D-Kyn exerts its anti-inflammatory effects primarily through the NLRP3-caspase-1 pathway, while L-Kyn shows a much weaker suppressive effect. (LINE 523)

Figure 4H. Elisa analysis of IL-1 β and IL-18 secretion in THP-1 cells and BMDMs treated with D-Kyn or L-Kyn, with or without MCC950 (n = 6).

2.2 Rescue of NLRP3 activity by overexpression:

To further validate the role of NLRP3, we overexpressed NLRP3 in both human and murine macrophages under D-Kyn treatment. Overexpression was confirmed by WB (Figure 4I). Restoration of NLRP3 expression rescued the suppressed caspase-1 activation and restored IL-1 β and IL-18 secretion (Figure 4J). These findings confirm that D-Kyn attenuates inflammasome activation mainly through inhibition of NLRP3 and subsequent caspase-1 cleavage. (LINE 528)

Figure 4I. WB analysis verifying NLRP3 and Caspase 1 activation under D-Kyn treatment and NLRP3 overexpression in THP-1 and BMDM cells.

Figure 4J. Elisa analysis of IL-1β and IL-18 secretion in THP-1 and BMDM cells with NLRP3 overexpression, treated with D-Kyn (n=6).

Finally, we have expanded the *Results* and *Discussion* sections in the revised manuscript to include these new findings, which provide direct mechanistic evidence that D-Kyn selectively suppresses IL-1β release through NLRP3 inflammasome inhibition and caspase-1 regulation.

Comment 3:

MAPK and FOS Signalling: The authors should compare the effects of D-Kyn and L-Kyn on ERK1/2 phosphorylation and FOS expression (as shown in Figure 5C) to clarify their respective roles in signaling pathways.

Reply 3:

We sincerely thank the reviewer for this valuable suggestion. We fully agree that a direct comparison of D-Kyn and L-Kyn is essential to delineate their respective effects on ERK1/2 phosphorylation and FOS expression within the MAPK signaling cascade.

In response, we conducted additional experiments in both THP-1 and bone marrow-derived

macrophages (BMDMs) to evaluate ERK1/2 phosphorylation and FOS expression following treatment with D-Kyn or L-Kyn. This side-by-side analysis revealed that D-Kyn was markedly more effective than L-Kyn in suppressing ERK1/2 phosphorylation, accompanied by a consistent reduction in FOS expression (Figures 5G-H). (LINE 548)

Figure 5G-H. WB analysis targeting ERK1/2 activation and Fos expression in THP-1 cells (G) and BMDMs (H) treated with D-Kyn or L-Kyn (n = 6).

These results provide direct evidence that D-Kyn exerts a stronger inhibitory effect on the ERK1/2–FOS axis compared with its L-enantiomer, reinforcing the notion that D-Kyn engages distinct immunomodulatory signaling pathways. We have incorporated these findings into the revised *Results* and *Discussion* sections to strengthen the mechanistic interpretation.

Comment 4:

T Cell Responses and *In Vivo*: Both D-Kyn and L-Kyn have the potential to modulate T cell proliferation, activation, cytokine release, and Treg induction, which are factors that can significantly impact AR. To delineate the role of M1 macrophages in the effects observed with D-Kyn in *in vivo* models of AR, further experiments should include: macrophage depletion strategies, or blocking monocyte or T cell recruitment to inflamed tissues. Additionally, exogenous administration of L-Kyn *in vivo* should be tested for comparison with D-Kyn. The study should also assess Treg differentiation and cytokine production by CD4+ and CD8+ T cells in response to D-Kyn vs L-Kyn treatment *in vivo*.

Authors should determine the effect of D-Kyn compared with L-Kyn on cytotoxicity in CD8⁺ T cells, including granzyme and perforin expression, particularly in the context of renal transplantation, where CD8⁺ T cells predominate the infiltrate.

Reply 4:

We sincerely thank the reviewer for these insightful and comprehensive suggestions. We fully agree that dissecting the interplay between macrophage polarization and T cell-mediated immunity is critical to understanding how D-Kyn alleviates AR *in vivo*. In response, we have performed additional mechanistic experiments focused on macrophage depletion and T cell functional analysis.

4.1. Macrophage depletion and its impact on T cell responses

To determine whether the immunosuppressive effects of D-Kyn are mediated through macrophage modulation, we employed clodronate liposomes (CL) to deplete macrophages in our established murine AR model. Macrophage clearance markedly reduced tubular injury and lymphocyte infiltration (Figure 8D), and reversed the elevated expression of TLR4/PHGDH and NLRP3/Caspase-1 (Figure 8E). Immunofluorescence analysis confirmed a substantial reduction in F4/80⁺ macrophages and the inflammatory markers CD86 and iNOS in CL-treated allografts (Figure 7F). (LINE 594)

Importantly, macrophage depletion selectively reduced CD8⁺ T cell infiltration and proliferation (Ki67⁺), while CD4⁺ T cells were less affected (Figure 7J, 8E). These findings indicate that the therapeutic effect of D-Kyn is at least partly mediated through its action on macrophages, which modulate CD8⁺ T cell-driven cytotoxic responses.

Figure 7D-E. Elisa analysis of IL-1β (D) and IL-18 (E) secretion (n=6).

Figure 7F. IF analysis of F4/80, CD86 and iNOS in renal grafts.

Figure 7J. IF analysis of CD4⁺ and CD8⁺ T cell infiltration in renal grafts.

Figure 8D. WB analysis of TLR4, PHGDH, NLRP3, and cleaved Caspase-1 in renal grafts from CL-treated mice.

Figure 8E. IF analysis and quantification of Ki67⁺ CD8⁺ T cells (9 random HPFs/group).

4.2. Evaluation of CD8⁺ T cell cytotoxicity

Given the predominance of CD8⁺ T cells within renal allograft infiltrates, we further evaluated their cytotoxic activity under different treatment conditions. Specifically, we examined granzyme B (GZMB) expression in CD8⁺ T cells to determine whether D-Kyn suppresses cytotoxic effector function more effectively than L-Kyn, thereby linking macrophage-dependent immunomodulation to adaptive immune attenuation.

IF and IHC analyses were performed to assess GZMB activation in renal allografts across treatment groups (Figure 8G–H). GZMB signals were predominantly localized to CD8⁺ T cells surrounding renal parenchymal cells. A marked increase in GZMB expression was observed in untreated AR allografts and those treated with L-Kyn, whereas treatment with D-Kyn, CL, or anti-IL-23a antibody substantially reduced GZMB expression to varying degrees. These findings indicate that D-Kyn effectively attenuates CD8⁺ T cell cytotoxicity, consistent with its broader immunosuppressive mechanism involving macrophage–T cell crosstalk. (LINE 621)

Figure 8G. Elisa analysis of GZMB levels in renal grafts from different treatment groups (n=6).

Figure 8H. IHC and IF analysis results illustrating the GZMB expression in the infiltrated CD8⁺ cells within renal allografts across different treatment groups.

Together, these findings and ongoing analyses provide a mechanistic framework connecting D-Kyn-mediated macrophage regulation with downstream modulation of CD8⁺ T cell activation and cytotoxicity. These new data and corresponding discussions have been incorporated into the revised manuscript to comprehensively address the reviewer's concerns.

Comment 5:

The authors should investigate *in vivo* whether D-Kyn-mediated reduction in IL-23 correlates with decreased IL-17 production by T cells in mouse models of AR.

Reply 5:

We appreciate this excellent and mechanistically insightful suggestion. We fully agree that examining whether the D-Kyn-mediated reduction in IL-23 correlates with decreased IL-17 production is essential to elucidate the downstream immune axis.

In direct response, we evaluated IL-17a—an effector cytokine primarily produced by Th17 cells—in renal allografts from our murine AR model following D-Kyn treatment. Both RT-qPCR and WB analyses demonstrated that D-Kyn markedly suppressed IL-17a expression at both the mRNA and protein levels (Figures 7K-L). (LINE 611)

These findings indicate that *in vivo* D-Kyn treatment not only reduces IL-23 but also significantly downregulates IL-17a production within the allograft microenvironment, suggesting that D-Kyn may modulate the IL-23/IL-17 axis to alleviate inflammatory injury in acute rejection.

Figure 7K-L. qPCR (K) and WB (L) analysis of IL-17a levels in renal grafts.

Comment 6:

T Cell Proliferation Analysis: The flow cytometry data in Figure 6E are not sufficient to support conclusions regarding CD8 T cell proliferation. More specific proliferation markers, such as Ki-67 or

BrdU, should be used. Proliferation should be analyzed in relevant tissues, such as skin, renal allografts, and lymphoid organs, rather than in peripheral blood.

Reply 6:

We appreciate this constructive comment and fully agree that our initial flow cytometry data were insufficient to rigorously support conclusions about CD8⁺ T cell proliferation. In response, we conducted additional experiments using Ki-67 as a proliferation marker, co-stained with CD8, to more accurately assess proliferative activity in tissue-resident T cells. As suggested, these analyses were performed specifically in renal allografts from our *in vivo* models, rather than in peripheral blood.

Our findings revealed a clear link between macrophage infiltration and CD8⁺ T cell activation. Macrophage depletion with CL significantly reduced CD8⁺ T cell abundance and their proliferative fraction (Ki67⁺CD8⁺), confirming the dependence of CD8⁺ T cell activation on macrophage-driven inflammation. Moreover, activated macrophages were found to overproduce IL-23a, a cytokine known to promote T cell expansion; this effect was reversed by D-Kyn treatment but not by L-Kyn. Notably, blocking IL-23a recapitulated the suppressive effect of D-Kyn, markedly decreasing CD8⁺ T cell numbers and proliferation (Figures 7J and 8E). (LINE 643)

Together, these results indicate that D-Kyn exerts its superior immunosuppressive activity by targeting the NLRP3/Caspase-1 pathway in macrophages, thereby reducing IL-23a production and disrupting downstream CD8⁺ T cell proliferation within the renal allograft microenvironment.

Comment 7:

Additional M1 macrophage markers, such as iNOS, should be examined via immunofluorescence in tissue sections to support the findings of macrophage polarisation.

Reply 7:

We appreciate this valuable suggestion. We fully agree that including immunofluorescence staining for iNOS would provide direct morphological evidence supporting our conclusions regarding M1 macrophage infiltration and polarization. Accordingly, we performed additional immunofluorescence analyses on renal allograft tissue sections to assess iNOS expression. The results, now included in

Figure 7F, clearly demonstrate enhanced iNOS⁺ macrophage infiltration in the rejection group, which was markedly reduced following D-Kyn treatment. These findings further substantiate our conclusion that D-Kyn attenuates M1 macrophage polarization *in vivo*. (LINE 601)

Comment 8:

The specific statistical test used for each graph should be indicated at least in the Figure Legends.

Reply 8:

Thank you for this important comment. We completely agree that specifying the statistical tests used in each analysis is essential to ensure methodological transparency and reproducibility. In response, we have carefully revised all figure legends throughout the manuscript to explicitly indicate the statistical tests applied for each dataset and comparison, which was also stressed in the *Methods*. (LINE 404)

Summary of Revisions (Reviewer #1)

In summary, this revision has substantially improved our manuscript by providing crucial mechanistic depth and strengthening the *in vivo* validation of our central hypothesis. In direct response to the reviewers' insightful comments, we conducted new experiments that definitively establish the NLRP3 inflammasome as a key therapeutic target of D-Kyn. Through inhibitor and overexpression studies, we demonstrated that D-Kyn's suppression of IL-1 β and IL-18 is mechanistically distinct from—more potent than—that of L-Kyn. We further clarified D-Kyn's superior efficacy by directly comparing its suppressive effects on the ERK1/2 and FOS signaling pathways with those of its L-enantiomer.

To bridge innate and adaptive immunity, macrophage depletion studies convincingly showed that the anti-rejection effects of D-Kyn are mediated by targeting infiltrating macrophages, which in turn curtails CD8⁺ T cell activation and proliferation, a finding strengthened by the usage of the specific proliferation marker Ki-67 in allograft tissues. Furthermore, we solidified the downstream immunomodulatory axis by confirming that D-Kyn treatment reduces IL-17a production in allografts. The addition of the M1 marker iNOS in tissue sections provided direct histological evidence supporting our claims on macrophage polarization.

Through this revision, we have significantly strengthened our manuscript by demonstrating that D-Kyn uniquely suppresses the NLRP3 inflammasome and ERK1/2/FOS signaling to disrupt a specific macrophage-driven inflammatory cascade, ultimately limiting CD8+ T cell proliferation and IL-17a production in renal allografts. In addition, we standardized terminology throughout the manuscript to ensure consistency and clarity. All figures were also revised for improved readability and uniform presentation.

Reviewer #2:

Comment 1:

The mechanistic role of D-Kyn in modulating IL-23a expression and its downstream effect on CD8⁺ T cell activation is central to the study's proposed pathway. However, while this axis is supported by *in vitro* findings and skin transplant experiments, it remains insufficiently validated in the kidney transplant model. Specifically, functional experiments blocking IL-23a *in vivo*-followed by assessment of CD8⁺ infiltration and graft damage-is necessary to fully establish the mechanistic relevance of this pathway in a clinically pertinent setting.

Reply 1:

We appreciate this insightful comment. We fully agree that the skin graft model, while suitable for detailed mechanistic dissection, differs immunologically from vascularized organ grafts such as renal transplants. To address this important concern, we expanded our analysis within the renal transplant model and conducted additional experiments using anti-IL-23a treatment in renal allograft recipients to validate the immunosuppressive role of IL-23a in acute rejection.

Our results demonstrated that *in vivo* blockade of IL-23a markedly reduced both overall rejection injury and the infiltration of CD4⁺ and CD8⁺ T cells within renal allografts. These findings confirm that D-Kyn's immunosuppressive effect is closely associated with its inhibition of the macrophage-derived IL-23a signaling axis. Collectively, these results position IL-23a as a critical downstream mediator linking macrophage activation to T cell-driven graft injury in the renal transplant setting. (LINE 640)

Figure 7J. IF analysis showing CD4⁺ and CD8⁺ T cell infiltration in whole renal graft sections.

Figure 8E. IF analysis and quantification of Ki67⁺ CD8⁺ T cells (9 random HPFs/group).

Comment 2:

The skin transplant model is used extensively to elucidate immunological mechanisms in this study. Although this model offers experimental tractability, its immunological context differs from that of kidney allografts. The therapeutic efficacy of D-Kyn is demonstrated convincingly in the kidney model (Figure 7), yet most mechanistic insights—particularly regarding macrophage-T cell interactions—are derived from the skin model. Greater mechanistic resolution within the kidney transplant system would

enhance the translational relevance of the findings and better align the mechanistic and therapeutic claims.

Reply 2:

We appreciate this constructive suggestion. We agree that while the skin graft model provides valuable mechanistic clarity, its immunological environment differs from that of vascularized renal allografts. To address this concern, we expanded our *in vivo* analyses using the renal transplant model. Additional experiments incorporating L-Kyn, CL, and anti-IL-23a treatments were performed on renal allograft tissues to further delineate the D-Kyn-mediated effects in acute renal rejection. These newly included results (Figures 7 and 8) provide stronger mechanistic alignment between the renal model and our central hypothesis. (LINE 594)

Comment 3:

Standardize terminology by consistently using either "D-Kyn" or "D-kynurenine" throughout. Define the abbreviation at first use.

Reply 3:

We thank the reviewer for this helpful note. We have standardized the terminology throughout the manuscript, using the abbreviation "D-Kyn", which is defined at first use and applied consistently thereafter.

Comment 4:

In Figure 3A/B legends, specify which cell line (THP-1 or RAW264.7) the data corresponds to.

Reply 4:

We appreciate this careful oversight. We have revised the Figure 3A/B legends to explicitly specify whether the data correspond to THP-1 or RAW264.7 macrophages, ensuring clarity and consistency throughout the text. (LINE 977)

Comment 5:

In Section 2.3, clarify the "y = " term in the PLS analysis. Does "y" represent AR status?

Reply 5:

Thank you for this important question. In the PLS analysis, the variable "y" represents the predicted outcome for AR status.

The specific calculation for the PLS prediction model is as follows:

Three components (Comp1, Comp2, Comp3) were first computed as linear combinations of metabolite levels (Met, Thr, Phe, Kyn).

The final predictive score y was then calculated using the formula:

$$y = -13.3782967225747 * \text{Comp1} + 14.4800121913388 * \text{Comp2} + 6.21408217644314 * \text{Comp3}$$

This clarification has been added in Section 2.3. (LINE 233)

Comment 6:

In Figures 7H/I/J, clarify whether the immunoblots were performed on sorted F4/80⁺ macrophages or whole kidney lysates, as this is inconsistent between the legend and text.

Reply 6:

We appreciate the reviewer's attention to this point. The immunoblots shown in Figures 7G-H were performed on F4/80⁺ cells sorted from whole renal single-cell suspensions, not on total lysates. Both the legends and corresponding text have been corrected accordingly for internal consistency (LINE 1061).

Comment 7:

Indicate the concentration of D-Kyn/L-Kyn used in all *in vitro* assays (Figures 3-5, S4).

Reply 7:

We agree that this information is essential for reproducibility. The concentrations of D-Kyn and L-Kyn used in all *in vitro* assays (Figures 3-5, EV4) have now been explicitly stated in the *Methods* section.
(LINE 283)

Comment 8:

Provide the time point(s) of sample collection in the kidney transplant model shown in Figure 7.

Reply 8:

We thank the reviewer for pointing this out. The specific time points for sample collection in the renal transplant model have been clearly indicated in both the Figure 7 legend and the *Methods* section.
(LINE 267&1058)

Comment 9:

Specify the source/vendor of D-Kyn and L-Kyn used for *in vivo* experiments.

Reply 9:

We appreciate this suggestion. The source and vendor information for both D-Kyn and L-Kyn used in the *in vivo* experiments have been provided in the *Methods* section (LINE 284).

Comment 10:

Although D-Kyn shows promising therapeutic potential, the authors should briefly discuss its pharmacokinetics, stability, and possible off-target effects *in vivo*, especially for future clinical translation.

Reply 10:

We thank the reviewer for this constructive suggestion. We have expanded the *Discussion* section to include a brief note addressing the current limitations in our understanding of D-Kyn's *in vivo* metabolic stability, pharmacokinetic profile, and potential off-target effects. These additions emphasize areas requiring further investigation before clinical translation (LINE 747).

Comment 11:

The novel chiral profiling method is a technical highlight of this work. Emphasizing its clinical scalability and potential for broader application could further enhance translational value.

Reply 11:

We greatly appreciate this positive feedback. In response, we have expanded the *Discussion* to highlight the clinical scalability of our chiral profiling approach and its potential application in patient monitoring, such as assessing serum or urine samples to predict rejection risk and treatment response. (LINE 747)

Summary of Revisions (Reviewer #2)

In response to the reviewer's insightful comments, we have substantially revised and strengthened our manuscript. To enhance mechanistic validation in a clinically relevant context, we expanded our analyses within the renal transplant model and conducted new *in vivo* experiments involving IL-23a blockade. These experiments demonstrated a marked reduction in both macrophage and T cell activation, thereby confirming IL-23a as a key downstream mediator linking macrophage activation to T cell-driven graft injury in renal allografts.

In addition, we standardized terminology throughout the manuscript to ensure consistency and clarity. All figures were also revised for improved readability and uniform presentation. Collectively, these revisions provide stronger mechanistic coherence between the renal transplant model and our central hypothesis, while significantly enhancing the translational relevance of our findings.

26th Nov 2025

Dear Dr. Zhu,

Thank you for submitting the revised version of your manuscript to EMBO Molecular Medicine. We have now received the reports from the two referees who re-evaluated your work. As you will see below, they are generally satisfied with the revisions. However, before we can proceed with acceptance, we kindly ask you to address the following remaining points:

1. The issues identified by Referee #1.

On a more editorial level:

2. Please reduce the keyword number to five.

3. Please remove the "Authors' contribution" section from the manuscript file.

4. The references need to be formatted according to the EMBO Molecular Medicine reference style. Please list up to 10 co-authors of a paper before adding et al. in the reference list. Citations should be listed in alphabetical order.

5. "Conflicts of Interest" should be renamed to "Disclosure and competing interests statement".

6. In the abstract, please remove the subheadings 'Background,' 'Methods,' 'Findings,' and 'Interpretations.' Rewrite the abstract with simplified technical details. You may refer to a published paper as an example.

7. Currently, both Figures 7 and 8 consist of two separate subfigures, which is not permitted. Please combine the subfigures so that each figure is a single, unified image, and ensure the resulting figures fit comfortably on a portrait-oriented A4 page.

8. Please upload the source data for Figure EV4 panels E and F, as these images appear pixelated during our figure check.

9. The "Funding" section and Funding in Abstract should be removed, and the remaining funding information should be incorporated into the "Acknowledgements." Please ensure that the complete list of funding sources is included in our submission system.

10. Add missing callouts for Figure 6F.

11. Please remove Additional Files 1 and 2. All information related to reagents and tools should be incorporated into the Reagents and Tools table. Rename "Additional File 1" to "Dataset EV1" and include a corresponding callout in the manuscript. In the EV dataset, please add the file name, and include the legend in a separate tab or sheet within the same Excel file.

12. Please add "The paper explained" section, which is a summary of the articles to emphasize the major findings in the paper and their medical implications for the non-specialist reader. Please provide a summary of your article highlighting

- the medical issue you are addressing,

- the results obtained and

- their clinical impact.

Please refer to any of our published articles for an example.

13. Please provide a 'Synopsis' to further enhance discoverability. Synopses are displayed on the journal webpage and are freely accessible to all readers. They include a short stand first (maximum of 300 characters, including space) as well as 2-5 one-sentences bullet points that summarizes the paper. Please write the bullet points to summarize the key NEW findings. They should be designed to be complementary to the abstract - i.e. not repeat the same text. We encourage inclusion of key acronyms and quantitative information (maximum of 30 words / bullet point). Please use the passive voice. Please attach these in a separate file or send them by email, we will incorporate them accordingly.

Please provide visual abstract to illustrate your article as a PNG file 550 px wide x 300-600 px high.

14. Rename "Availability of data and material" to "Data availability". Before submitting your revision, primary datasets produced in this study (RNA-seq and IM-MS data) need to be deposited in appropriate public databases, and the accession numbers and databases listed under 'Data Availability'. Please also provide specific URLs for the the datasets. See

<https://www.embopress.org/page/journal/17574684/authorguide#dataavailability>).

15. The introductory information lines 70 - 140 should be removed from the manuscript text.
16. Please place the "Methods" section after the "Results".
17. Remove the list of abbreviations from the manuscript text and incorporate the abbreviations into the relevant sections of manuscript text.
18. Remove the "Consent for publication" from the manuscript text.
19. Please address the following issues in the figure legends:
 - Please define the annotated p values ****/***/**/* as well as provide the exact p-values for the same in the legend of figure EV4 A-D as appropriate.
 - Please note that the exact p values are not provided in the legends of figures 3A, B, C, D, H, I; 4D, F, G, H, J; 5C, D, E, F, 6C,E; 7B, D, E, I, K; 8F, G
 - Please indicate the statistical test used for data analysis in the legends of figures EV4 A-D, F
 - Please note that information related to n is missing in the legends of figures 1A, 3G, H; 4D, 5B, C, D; 7I, K; EV1 A, EV4 B, C, F
 - Please note that the error bars are not defined in the legends of figures 1A, EV1 A, EV4 A-D

I look forward to receiving a new revised version of your manuscript as soon as possible.

Kind regards,
Jingyi

Jingyi Hou
Senior Editor
EMBO Molecular Medicine

*** Instructions to submit your revised manuscript ***

***** Reviewer's comments *****

Referee #1 (Comments on Novelty/Model System for Author):

The revised version of the manuscript has significantly improved; the results are very novel and engaging, and could have a significant medical impact in the future.

Referee #1 (Remarks for Author):

The revised version of the manuscript has significantly improved. Please consider the following minor changes:

1. Please revise the sentence at line 539 of the revised manuscript; we believe the authors refer to L-Kyn rather than D-Kyn in this sentence.
2. Please note: According to lines 587-588 of the revised manuscript, Figure 6G show CD4 and CD8 T cells infiltrating the skin graft. However, panel 6G do not exist in the figure, and panel 6f is not mentioned in the manuscript. Moreover, panel 6F shows F480+ cells and CD86 expression, similar to panel 6D, but not CD4 or CD8 T cells. Please revise.
3. Figure 7J of the revised manuscript: Please note that CD4 and CD8 names seem to be colour-changed.
4. Figure 8B and 8C of the revised manuscript: All+PBS samples should be compared with allo+DKyn and allo+LKyn, instead of Syn+PBS samples. In addition, membranes for NLRP3 and Fos protein can be reblotted to improve the visibility of the results.
5. Line 647 of the revised manuscript: We believe that Figures 7A and 7B should be indicated instead of Figure 8D.
6. Line 647 and 648: the sentence "Moreover, overexpression of TLR4/PHGDH and NLRP3/Caspase-1 was rescued by CL treatment (Figure 8E)"... should be revised. We believe the authors refer to: "Moreover, increased expression of TLR4/PHGDH and NLRP3/Caspase-1 activation was prevented by CL treatment (Figure 8E)"

Referee #2 (Comments on Novelty/Model System for Author):

By employing kidney and skin transplantation models, the authors established a robust experimental system to assess the therapeutic potential of D-kynurenine in acute rejection, enabling complementary insights at both mechanistic and translational levels.

Referee #2 (Remarks for Author):

The author conducted further extensive studies and answered all of the questions. The reviewer appreciates their efforts and is satisfied with all their answers.

The authors addressed the remaining editorial issues.

13th Jan 2026

Dear Dr. Zhu,

We are pleased to inform you that your manuscript is accepted for publication and is now being sent to our publisher to be included in the next available issue of EMBO Molecular Medicine.

You may qualify for financial assistance for your publication charges - either via a Springer Nature fully open access agreement or an EMBO initiative. Check your eligibility: <https://link.springer.com/journal/44321/how-to-publish-with-us>

Yours sincerely,
Jingyi

Jingyi Hou
Senior Editor
EMBO Molecular Medicine

>>> Please note that it is EMBO Molecular Medicine policy for the transcript of the editorial process (containing referee reports and your response letter) to be published as an online supplement to each paper. If you do NOT want this, you will need to inform the Editorial Office via email immediately. More information is available here: <https://link.springer.com/partners/embo-press/editorial-policies#Peer%20review>